# A Survey on LLM Test-Time Compute via Search: Tasks, LLM Profiling, Search Algorithms, and Relevant Frameworks

**Xinzhe Li** *sergioli212@outlook.com*

**Reviewed on OpenReview:** *https://openreview.net/forum?id=x9VQFjtOPS*

## Abstract

LLM test-time compute (or LLM inference) via search has emerged as a promising research area with rapid developments. However, current frameworks often adopt distinct perspectives on three key aspects: task definition, LLM profiling, and search procedures, making direct comparisons challenging. Moreover, the search algorithms employed often diverge from standard implementations, and their specific characteristics are not thoroughly specified. This survey aims to provide a comprehensive but integrated technical review on existing LIS frameworks. Specifically, we unify task definitions under Markov Decision Process (MDP) and provides modular definitions of LLM profiling and search procedures. The definitions enable precise comparisons of various LLM inference frameworks while highlighting their departures from conventional search algorithms. We also discuss the applicability, performance, and efficiency of these methods. For ongoing paper updates, please refer to our GitHub repository: `https://github.com/xinzhel/LLM-Search`.

## 1 Introduction

Auto-regressive Large Language Models (LLMs) such as GPT-4 (Achiam et al., 2023) have achieved impressive performance across a range of tasks using few-shot prompting (Radford et al., 2019; Brown et al., 2020) or even zero-shot prompting (Kojima et al., 2022). They demonstrate broad world knowledge, commonsense, and human-like reasoning for problem solving (Santurkar et al., 2023; Wang et al., 2022; Zhong et al., 2022; 2023). Nonetheless, these models still struggle with complex tasks that require long-horizon planning and deep reasoning. This limitation motivates extending sequential decoding with search methods (Yao et al., 2023a; Hao et al., 2023), an approach we refer to as **LLM Inference via Search** (**LIS**) in this survey. [1]

To clarify the specific reasoning and planning tasks that challenge LLMs, we distinguish between two general categories:

1. **Language Reasoning Tasks**: These tasks align closely with typical Natural Language Processing (NLP) problems but require multi-step reasoning, such as solving math word problems (Cobbe et al., 2021). Some prior works classify language reasoning as a form of planning (Huang et al., 2024). Anyway, LLMs frequently achieve near-saturation performance on these tasks, especially when explicitly prompted to generate intermediate steps using methods like Chain-of-Thought (CoT) (Wei et al., 2022). Valmeekam et al. (2023a) argue that the reason for such high performance is that the solutions and reasoning paths for these tasks are likely included as world knowledge and commonsense in the LLM training corpus. To more rigorously evaluate reasoning capabilities, Valmeekam et al. (2023a) introduce problem sets that involve multi-step reasoning beyond traditional NLP contexts, leading to the second category.

2. **Non-NLP Tasks:** These tasks typically involve simple decision-making environments (such as games) and dynamic real-world scenarios (such as household or logistics tasks) traditionally addressed by reinforcement learning and classical planning algorithms. Current state-of-the-art LLMs

---

[1] While related, frameworks discussed in Section 6 are not categorized under LIS.

show significantly lower performance compared to humans on benchmarks in these domains, including PlanBench (Valmeekam et al., 2023a;b) and SearchBench (Borazjanizadeh et al., 2024).

Importantly, our survey not only addresses the challenging scenarios of planning but also systematically explores all tasks that can be formulated within LIS frameworks, including language reasoning, sequential decision-making tasks like robotics (Putta et al., 2024) and graph-traversal tasks such as path finding (Meng et al., 2024). The following two subsections highlight the unique perspectives and novel contributions offered by this survey.

## 1.1 Existing Surveys

Current reviews on LLM search are limited from the following perspectives.

**No dedicated, Detailed Survey**  Current surveys only contain paragraphs or sections to roughly touch on both technical aspects and their practical applicability, as summarized in Table 1.

**Limited Mention on LLM-Side Design**  Specifically, most of existing surveys (Huang et al., 2024; Wang et al., 2024b) mention little or a few implementations and dimensions for LLM profiling, which is not suitable for all the frameworks. Besides, the lack of examples hinders understanding.

**Limited Mention on Search**  Li (2024) give more detail regarding LLM profiling but lack details on search processes. Nonthelessness, most of existing surveys (Huang et al., 2024; Wang et al., 2024b) give a general sense of the computation process by mentioning which classical search algorithms the frameworks are built upon (e.g., depth-first search). However, details should be given because of their nuanced differences. Besides, many untypical twists to classic search algorithms are hidden. The deviations are not friendly for newcomers in the newly-developed area, e.g., those computer science graduates educated with typical search algorithms.

Table 1: Comparisons with other surveys related to LLM inference via search (LIS). **"Coverage"** indicates the number of related papers, **"Sections"** refers to the sections of the survey manuscripts that discuss LLM inference via search, and **"Def."** is the abbreviation for Definition. The URLs for the reference versions are given at the footnotes. **"Mentioned"** refers to the mention of the LLM function, while **"Limited"** means that a formal definition is given without specific distinctions. The numerical values correspond to the number of implementations (**impl.**) or dimensions (**dim.**) identified for LLM-Profiled Roles (LMPRs); different dimensions may lead to a combinatorial number of implementations. **"25+"** represents the number of papers detailed in Sections 5 and 6, excluding those used solely for benchmarking, analysis, or from related domains.

| | Coverage | Sections | Task Def. | Search Algorithms | | LLM Profiling | | |
| | | | | Details | Deviations | Policy | Value | Transition |
| --- | --- | --- | --- | --- | --- | --- | --- | --- |
| Huang et al. (2024) | 5 | ✓(§4) | ✗ | ✗ | ✗ | 1 impl. | 1 impl. | ✗ |
| Wang et al. (2024b) | 3 | ✓(§2.1.3) | ✗ | ✗ | ✗ | ✗ | ✗ | Mentioned |
| Li (2024) | 8 | ✓(§4.3) | ✗ | Limited | ✗ | 2 impl. | 2 dim. | NA |
| Ours | 25+ | All | ✓ | ✓ | ✓ | 8 impl. | 4 dim.; 14 impl. | 2 impl. |

## 1.2 Survey Structure

To solve the above limitations, we provide unified **task definitions** and decouple the **LLM-specific design (mainly prompting)** from **the control program (search procedures/algorithms)**. There exists a hierarchical structure between them: the low-level definitions provide a unified interface for the high-level components. The overall structure, accompanied by illustrative examples, is presented in Figure 1.

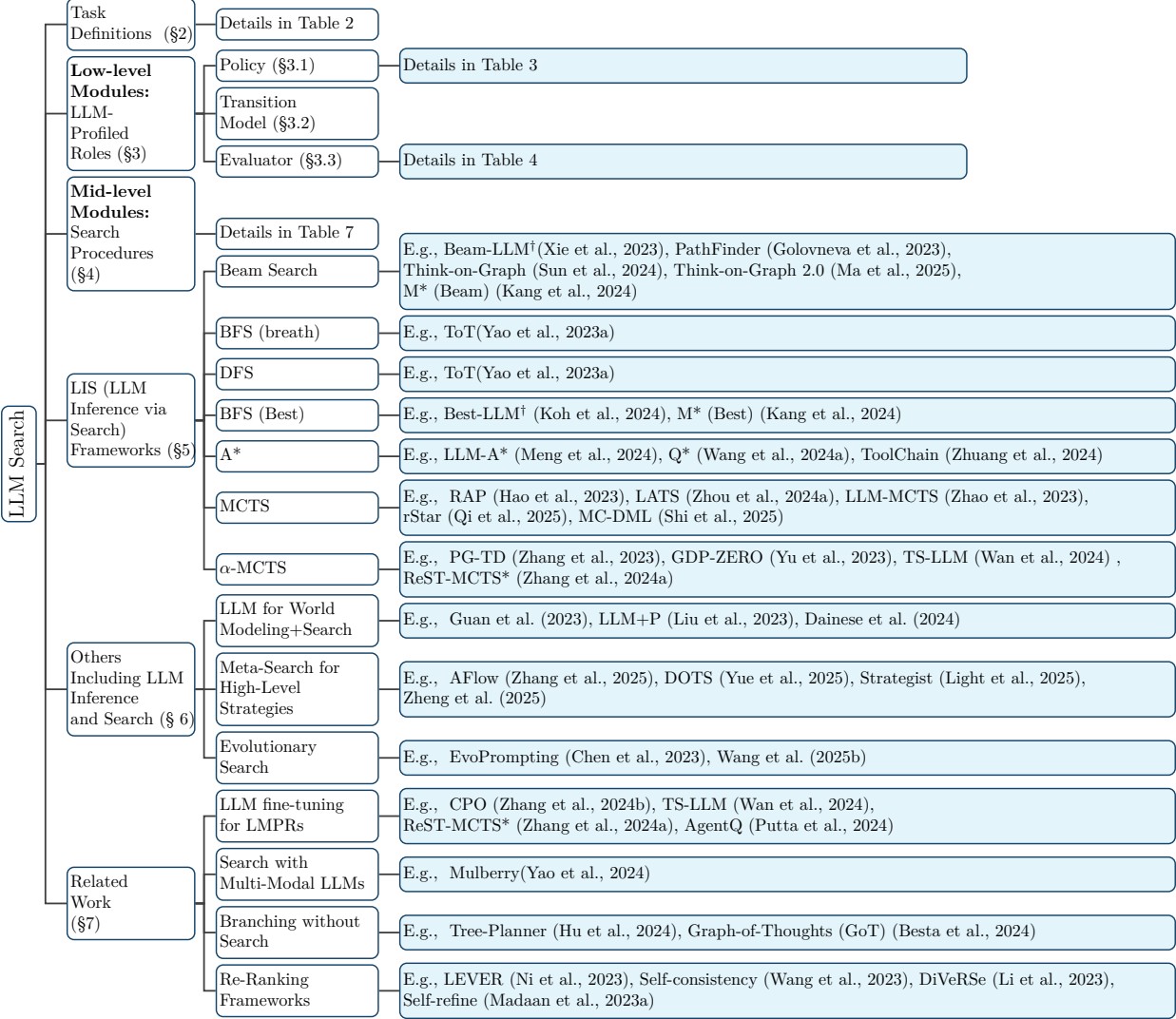

Figure 1: Survey structure. This table shows a selected subset of the works reviewed. Here are two notes: 1) To avoid duplication, comprehensive lists of related work for **Task Definitions** and **LLM-Profiled Roles** are provided in the refered tables; 2) Framework names marked with a sword symbol ($^\dagger$) denote those devised by the authors of this survey, rather than being directly drawn from existing literature.

**Introducing a Unified Task Definition Based on MDPs** (§ 2) Our definition standardizes different tasks in MDP structure. While MDPs naturally align with AI domains like robotics, special attention is given to adapting this definition for tasks traditionally not modeled as MDPs, such as graph traversal, reasoning, dialogue systems, and code generation. Notably, this MDP-based definition is also applicable to other LLM inference frameworks beyond search, including works like Li et al. (2022), Zhao et al. (2023), and Hu et al. (2024).

**Comprehensively Summarizing LLM Profiling and Implementations** (§ 3) The design and implementation of LLM profiling and prompting can be modularized into components commonly used in solving MDPs (Sutton & Barto, 2018): policies, value functions, and transition models. Correspondingly, 3 types of LLM-Profiled Roles (LMPRs) are defined.

**Defining Modular Search Procedures** (§ 4) Rather than directly showcasing individual search-based frameworks for LLM inference, we focus on modular and reusable components to reduce redundancy and enable more straightforward comparisons across frameworks. This approach promotes flexibility and minimizes overhead when adapting or extending search methods.

**Reviewing Individual Frameworks** (§ 5) Based on the unified task and LMPR interface, we provide a comprehensive review of individual frameworks, organized by the search algorithms they are built upon. Our analysis highlights how LLM integration either diverges from or enhances traditional search algorithms. We identify and clearly present 11 frameworks, summarized in Table 7. This count exclusively includes frameworks that focus on test-time computation through search detailed in Section 5. Note that our implementations may differ from the specific approach described in the original manuscript, as our goal is to provide a unified interface that facilitates easier comparison among various methods. Nevertheless, they are functionally equivalent. For example, whereas Koh et al. (2024) implement Best-first search using a priority queue, we achieve the same outcome by employing a top-k selection procedure that is common to MCTS and other search algorithms.

**Comparisons with Other Test-Time Frameworks** Additionally, we highlight other test-time frameworks that function as components within search processes, such as ReAct (Yao et al., 2023b), CoT (Wei et al., 2022), and Self-Consistency (Wang et al., 2023), along with those discussed in Section 7.

**Analyzing Key Perspectives of LIS Frameworks** (§ 8) We critically examine these methods from four key perspectives: **deviation**, **applicability**, **performance**, and **efficiency**. For deviations, we compare the structural and functional differences between search procedures in LIS frameworks and standard search algorithms as described in foundational texts such as Russell & Norvig (2010) and Sutton & Barto (2018). This analysis highlights how LLMs modify or enhance traditional search processes, providing a deeper understanding of their impact and potential.

**Other LLM Inference + Search Directions** (§ 6) This survey primarily focuses on LLM inference via search, where downstream tasks are formulated as sequential decision-making problems, and LLMs serve as integral components. This focus allows us to present a detailed, concise, and systematic analysis. However, confining our discussion solely to these frameworks might give the impression that the title overstates our scope. Hence, Section 6 covers additional research directions that also involve LLM inference and search.

## 1.3 Intended Audience and Use Cases

**Reviewed Venues** Our goal is to provide a comprehensive overview of the latest research from active venues. To this end, we systematically search high-quality conferences that publish work on LLM inference via search, including ICLR, ICML, and NeurIPS from 2023 to 2025[2], and we pay special attention to ACL, EMNLP, and NAACL conferences. We acknowledge, however, that the rapid pace of advancements in the field means that some forthcoming papers (that are under review or available as preprints) may not be covered. Nonetheless, this survey offers a curated collection of classical and reusable implementations that serve as a solid foundation for future research and development.

**Support for Reviewers and Researchers** This survey is intended to help reviewers and researchers more effectively assess the novelty of their contributions relative to prior work. After reviewing papers and evaluating the reviews of papers on OpenReview,, we have observed that some claims of novelty in existing research may be inadvertently inaccurate due to the rapid pace of developments and the entanglement of

---

[2]Excluding ICML2025 and NeurIPS2025, which have not yet been released.

task novelty and technical novelty in this area. Researchers and reviewers from the venues we specify above might have additional interests.

**For Research Engineers** To support practical implementation, search procedures (Section 4) are presented in an Object-Oriented Programming (OOP) style, promoting modularity and ease of integration for research engineers.

**Anchoring Purpose** Our work serves as a valuable reference in two key ways: **1) Incorporating Novel Designs with Minimal Adjustments**: During the development of this survey, we seamlessly integrated emerging methods with minor modifications. For example, we separated single-step simulation from path simulation in MCTS to accommodate LATS (Zhou et al., 2024a) and decoupled value-based selection from LMPE+ or state-based evaluation to allow specialized ways to assign values, e.g., using parents' states for evaluation Koh et al. (2024). **2) Expanding with Additional Details**: Our structured presentation of search control flows serves as a stable anchor for integrating such complex components. Although many of these design elements are non-trivial, our framework simplifies readers' understanding of their integration by using existing search process as a guiding anchor.

### 1.4 Limitations

We adopt Markov Decision Process (MDP) definitions to unify and compare various methods due to their comprehensive nature. However, this formalism may feel excessive for readers focused on specific frameworks or tasks where transition and action definitions are unnecessary. For instance, the Tree-of-Thoughts (Yao et al., 2023a) approach could be more intuitively understood without relying on MDP foundations.

## 2 Task (Re)formulation

Tasks solved by LLM-integrated search are inherited from both the "LLM" side (human language tasks) and the "search" side (structured tasks): 1) language reasoning: LLMs are naturally applied to reasoning tasks in language (Wei et al., 2022). 2) structured tasks: On the other hand, search algorithms are more conventionally utilized for structured tasks, such as web navigation, robotic navigation, gaming, and graph traversal (Russell & Norvig, 2010; Sutton & Barto, 2018). The convergent nature is that all of them belongs to sequential decision-making, e.g., reasoning often involves generating and evaluating sequences of logical steps or decisions to arrive at a conclusion.

**A MDP-Like Formulation** To enable a clear comparison across different frameworks, this section formulates the tasks in Markov Decision Processes (MDPs) $\langle S, A, \mathcal{T}, R \rangle$. In addition, observations $O$ are often considered in Partially Observable Markov Decision Processes (POMDPs) (Li et al., 2022; Zhao et al., 2023; Hu et al., 2024):

- **A set of states** $S$, including the goal state $s_g$;

- **A set of observations** $O$, where $o_t$ is the partial observations of the state $s_t$ at time step $t$;

- **A set of actions** $A$, where $a_t \in A$ is the action on $s_t$;

- **Transitions** $\mathcal{T}(s_{t+1} \mid s_t, a_t)$, which define the dynamics from one state to another after executing an action;

- **Rewards** $R$: A rewards evaluate the "quality" of a state or a trajectory towards the desired outcomes or goals. They can be delivered in two ways: **outcome rewards** are provided at the end of an episode to assess the overall quality of the final state, whereas **process rewards** are given throughout the episode to reflect the quality of the agent's behavior as it progresses.

Before introducing concrete tasks in the subsections, some necessary highlights can facilitate the understanding:

Table 2: A unification of various tasks for LLM Inference via Search (LIS). A single benchmark may be configured for different tasks. Therefore, the "Benchmark" column specifies each task's benchmark alongside the corresponding adapting frameworks *(listed in parentheses)*.

| Tasks | Actions | States / Observations | Transitions | Rewards | Action Reversible? | Benchmarks (*Used by*) |
|---|---|---|---|---|---|---|
| Embodied Tasks | Discrete, constrained, heterogeneous, state-dependent | Discrete | Normally deterministic | $R_{\text{default}}$: 1 if the goal is achieved | Maybe | VirtualHome (Puig et al., 2018) *(Tree-Planner)*, Jericho (Hausknecht et al., 2020) *(MC-DML)* |
| Combinatorial Tasks | Discrete, constrained, state-dependent | Discrete | Deterministic | $R_{\text{default}}$ | Maybe | Game-of-24 (Yao et al., 2023a) *(ToT, TS-LLM)*, Chess (Wan et al., 2024) *(TS-LLM)* |
| Web Navigations | Discrete, constrained, heterogeneous | Discrete, infinite | Dynamic or Deterministic | $R_{\text{default}}$ | Maybe | WebShop (Yao et al., 2022) *(Agent Q, LATS)*, WebArena (Zhou et al., 2024b) *(Best-LLM)* |
| Graph traversal | Discrete, constrained, homogeneous | Discrete, finite (commonly) | Deterministic | 1 if $s \in G$ | Maybe | GridMap (Meng et al., 2024) *(LLM-A\*)* |
| Reasoning ($\mathcal{T}$ via Concatenation) | Open (A thought expressed in one or more tokens) | Open, unknown until reached (Problem description + concatenated thoughts) | Deterministic (Concatenating) | 1 if the final results = ground-truth | ✓ | GSM8K Cobbe et al. (2021) *(Q\*, M\*, ReST-MCTS\*)*, Math (Hendrycks et al., 2021b) *(Q\*, ReST-MCTS\*)*, HotpotQA (Yang et al., 2018) *(CPO, rStar)*, ToT-Writing (Yao et al., 2023a) *(ToT)* |
| Reasoning via QAs | Open | Open, unknown until reached | Dynamic (Question answering then concatenating Q&A) | 1 if the final results = ground-truth | ✓ | HotpotQA (Yang et al., 2018) *(rStar)*, GSM8K Cobbe et al. (2021) *(RAP)* |
| Reasoning ($\mathcal{T}$ via concatenations and tool invocation) | Open | Open, unknown until reached | Deterministic (Concatenating); dynamic or deterministic (tool invocation) | 1 if the final results = ground-truth | Maybe | HotpotQA (Yang et al., 2018) *(LATS)* |
| Reasoning Over Knowledge Graph | Discrete, constrained, heterogeneous | Open thoughts + discrete and finite entity-relation triplets | Deterministic (Concatenating) | 1 if the final results = ground-truth | ✓ | WebQSP (Yih et al., 2016) *(Think-on-Graph, Think-on-Graph 2.0)* |
| Tool-based tasks | Discrete, constrained, heterogeneous | Problem description + concatenated actions | Deterministic (Concatenating) | 1 if the task is completed | Maybe | GSM8K Cobbe et al. (2021) *(ToolChain\*)*, ToolBench (Xu et al., 2023) *(ToolChain\*)* |
| Code Generation | Open (A single token) | Open, unknown until reached (Problem description + concatenated tokens) | Deterministic | E.g., pass rate of the complete program | ✓ | MBPP (Austin et al., 2021) *(LATS, Q\*)*, APPS (Hendrycks et al., 2021a) *(PG-TD)* |
| Goal-oriented Dialog | Discrete, constrained (An intent) | A sequence of intents, agent/user utterances | Dynamic | 1 if the conversational goal is achieved | ✗ | PersuasionForGood (Wang et al., 2019) *(GDP-Zero)* |

- **Tasks (this section) vs. search agents (The following sections)**: This section only discuss task elements that are external to agents and exist independently of how agents operate or learn.

We will see in the next section how the POMDP setting fits in LMPRs and agent definitions, where the Markovian assumption is broke.

- **MDPs for both concrete and abstract domains**: MDP settings are purely conceptual. They do not need to mirror real-world states such as sensor readings, a robot's position, or a digital image snapshot. They also do not have to replicate actual actions like a robot arm lifting an object, a self-driving car navigating intersections, or selecting a digital advertisement. Similarly, the feedback mechanisms for reward shaping can differ from those in the real world, such as a thermometer's continuous reading or a binary success/failure indicator. Therefore, we can categorize the tasks into concrete and abstract domains:

    - Concrete domains: In tasks like recycling robot, gridworld, and chess, actions and states are always naturally defined. They are always modeled as MDPs in the study of Reinforcement Learning (RL) (Sutton & Barto, 2018), intelligent agents, and control theory. Commonly, a physical environment or a rule set defines a discrete and finite action space. And states are commonly finite and can be enumerated, e.g., all the possible configurations of chess board or the grid areas robot can travel.
    - Abstract domains: However, others, e.g., graph traversal and reasoning tasks, are not commonly formalized as MDPs until the emergent of LLM-based agents. In particular, the MDP elements in graph traversal and reasoning tasks are not explicit. The rest of this section mainly discusses how the these tasks fit the following MDP notations, as summarized in Table 2.

We highlight this to prevent readers from unnecessary confusion because some task settings seem counterintuitive.

## 2.1 Embodied Tasks in Text

Embodied tasks involve simulating physical interactions and spatial relationships analogous to those in the real world. In these tasks, agents must navigate through environments, manipulate objects, and perform other complex physical activities that result in observable changes. The simulated environments can range from realistic household settings (e.g., AlfWorld (Shridhar et al., 2021), VirtualHome (Puig et al., 2018)) to realms inspired by Interactive Fiction (IF) games (e.g., Minecraft (Fan et al., 2022), Jericho (Hausknecht et al., 2020)).

- **States**: A state captures the current physical configuration of the environment, detailing the positions and conditions of both objects and the agent.

- **Actions**: The action space is dynamic and varies based on the current state of the environment. While identifying valid actions can be part of the agent's job, existing benchmarks often provide a predefined set of valid actions for each state.

- **Transitions**: State transitions $\mathcal{T}$ represent the physical changes occurring in the environment, which are typically discernible through common sense in a single-player setting.

- **Default Rewards**: Rewards are typically assigned when the agent reaches a goal state $g$. The default reward function $R_{\text{default}}(s)$ follows a binary scheme: $R(s) = 1$ if $s \in G$ (i.e., $s$ is a goal state), and $R(s) = 0$ otherwise.

## 2.2 Combinatorial Tasks in Text

Combinatorial tasks focus on strategic reasoning and decision-making within abstract, rule-governed environments. In these tasks, agents must explore discrete state spaces, evaluate possible moves, and plan sequences of actions to achieve a desired outcome. The environments are defined by clear, deterministic rules and often involve well-known games or puzzles, such as chess and the game of 24.

- **States**: A state encapsulates the current configuration of the game or puzzle. For example, in chess, the state is defined by the positions of all the pieces on the board, while in the game of 24, it represents the current arrangement of numbers and operations available.

- **Actions**: Actions are the discrete moves or operations that alter the state. In chess, these are the legal moves of the pieces, and in the game of 24, they are the mathematical operations (addition, subtraction, multiplication, division) used to combine numbers.

- **Transitions**: Transitions describe the evolution of the state following an action, adhering strictly to the game's rules. These transitions are deterministic, meaning that the same action in a given state will always yield the same new state, as seen in the systematic progressions in chess or the arithmetic rules in the game of 24.

- **Default Rewards**: Rewards are typically associated with achieving specific goals, such as winning a game or solving the puzzle. For instance, in chess, a reward might be assigned upon checkmate, whereas in the game of 24, the reward is granted when the target number is correctly reached.

## 2.3 Web Navigation

Another type of tasks is to navigate on websites for shopping and retrieving information (Zhou et al., 2024b)

- **States/observations**: A state/observation is normally a web page. For example, the beginning state can be the homepage. The transition is governed by a deterministic transition function. The state is not always accessible to the agent. For example, "$s_t$ may include private information such as database entries of the site" (Koh et al., 2024)).

- **Transitions**: Transitions occur when the agent interacts with webpage elements—such as clicking links, filling forms, or selecting menu options—to move from one page (state) to another. These transitions are governed by a deterministic function, meaning that a specific action reliably leads to the same subsequent state, although dynamic content or session-specific data might occasionally influence the exact presentation of the resulting page.

- **Reward**: A default reward function, $R_{\text{default}}$, is frequently employed; for example, a goal state might be defined as successfully ordering a desired product on a website.

## 2.4 Graph Traversal in MDPs

A graph traversal problem, e.g., robotic navigation, is represented as a graph $G = (V, E)$, where $V$ is a set of vertices (or nodes), and $E \subseteq V \times V$ represents the transitions between them. However, this definition is the common task settings for uninformed and informed search. However, these algorithms integrated with LLMs can be generalized beyond this definition. This is why recent work (Yao et al., 2023a) that uses these search algorithms along with LLM often uses MDP terminology, but does not include formal unification. Hence, we propose a conceptual framework that views graph traversal as a simplified, deterministic form of an MDP, where actions and transitions are predefined by the graph structure. This framework can be described as follows:

- **States**: Each node $v \in V$ is a state.

- **Actions**: Actions are represented by edges $E$. The action space is generally considered homogeneous because the type of action is uniform, such as "MOVE[arg]", where "[arg]" represents parameters like direction or target node.

- Transitions: Following an edge from a node $s$ via an action (edge) $(s, s') \in E$ always leads to the same node $s'$. Hence, the transitions $\mathcal{T}$ are deterministic.

- **Rewards**: Besides the default one, an estimated cost-to-go from node $s$ to a goal node $g \in G$ can be defined for process rewards in the MDP formulation. This is explicitly defined in A* as a heuristic guiding the search

As best as we know, this conceptualization is not explicitly stated in any peer-reviewed literature.

## 2.5 Language Reasoning in MDP

The formulations of language reasoning tasks are more diverse and creative. Although not exhaustive, the following paragraphs summarize forms that are particularly used in the current study of LLM-integrated search.

**Reasoning ($\mathcal{T}$ via Concatenation)**  A reasoning process can be concretized as a Chain of Thoughts (CoT) $T_1, T_2, \ldots$ (Wei et al., 2022), each expressed as a sequence of tokens via LLM generation. Reasoning steps can evolve naturally, ranging from a single word to a full sentence (Zhang et al., 2024a). However, ensuring clear semantics for each step remains a challenge. Alternatively, these steps can be explicitly defined. For instance, in creative writing, the first step can be specified as planning, and the second step is to generate according to the plan (Yao et al., 2023b). Following previous work (Li, 2024; Wang et al., 2024a), the MDP formulation includes:

- **Actions**: An action is a thought consisting of several tokens, i.e., $a_1 = T_1$.

- **States**: The initial state $s_1$ is defined by the task information, e.g., a user query, a problem description or the goal. The following states are defined as the concatenation of the following thoughts:

$$s_t = (s_{t-1}, a_{t-1}) = (s_1, a_1, \ldots, a_{t-1}) \tag{1}$$

   Apparently, directly concatnenating open actions leads to the open state space. When the reasoning is naturally evolved, the final state $s_g$ comes when the final thought $T_g$ or the entire chain expresses a valid response. It can be known in which step $s_g$ is reached for deliberate reasoning steps.

- **Transition**: The deterministic state transition is defined for reasoning tasks. The next state $s_{t+1}$ is equal to the concatenation of $s_t$ and $a_t$.

- **Rewards**: An outcome reward is given when the final answer matches the ground truth and fits in human preference.

**Reasoning via QAs**  Some other works deliberately formulate an action space. A given task is decomposed into sequentially dependent subtasks (actions) requiring an Execute function, which relies on LLMs to solve. In other words, LLMs can be considered as transitions. Recent work on LLM search formulates a subtask as a question and use LLMs to answer the question. These subtasks (i.e., actions) can be generated all at once (Zhou et al., 2023) or in sequential order (Hao et al., 2023), each can be defined as an action $a_t$. $s_t$ is the concatenation of the task information $s_0$ and all the questions already answered with their answers: $question_1, answer_1, \ldots, question_t, answer_t$.

**Reasoning ($\mathcal{T}$ via Concatenation and Tool Invocations)**  Once tool definitions are given to LLMs. During reasoning, LLMs can generate specific tokens to invoke tools. This integration of tool invocation into reasoning is firstly proposed by ReAct Yao et al. (2023b) and recently adapted to the LIS framework (Zhou et al., 2024a). Based on the definitions for *Reasoning ($\mathcal{T}$ via Concatenation)*, the following things are added:

- **Actions**: Tool-related actions are commonly discrete and constrained. For example, when Wikipedia is used, the possible actions include search[arg], lookup[arg]. Although not always, most works on LLM tool use only include the reading-only actions, the actions are reversible.

- **Transitions**: Due to the change of the action space, the transitions can be either dynamic or deterministic, depending on the tools.

- **States**: a state now concatenates not only the LLM generation but also tool responses. Moreover, LLM generation is not only about the direct thoughts for tasks but also the actions for tool invocations.

$$s_t = (s_1, a_1, o_1, \ldots, a_{t-1}, o_{t-1}) \tag{2}$$

**Reasoning Over Knowledge Graph**   Before defining reasoning over a knowledge graph, we clarify why this task is distinct from both graph traversal and reasoning under tool invocations.

- **Not a Tool Invocation:** In reasoning under tool invocations, LLMs autonomously decide whether to use external tools. In contrast, a knowledge graph is an integral part of the task environment on which the LLM agent operates.

- **Different from Graph Traversal:** In typical graph traversal tasks, the graph directly models a visible or observable world. A knowledge graph, however, is a carefully designed structure that captures heterogeneous semantic relationships between entities or concepts.

We now define the task of reasoning over a knowledge graph:

- **States**: In graph traversal, each node $v \in V$ represents a state. In this context, the state is composed of all explored nodes (entities) and their interrelationships, which collectively inform subsequent decisions and the final outcome. Additionally, any relevant LLM-generated knowledge that contributes to these decisions is incorporated into the state.

- **Actions**: The action space consists of exploring new relations and entities within the knowledge graph, as well as generating new knowledge through LLM outputs.

- **Transitions**: A transition deterministically concatenates the new information (e.g., a discovered relation or entity) with the existing state, reflecting the edge $(s, s') \in E$ between the current state and the new information.

- **Rewards**: $R_{\text{default}}$ is used by default.

## 2.6   Tool-Based Tasks in MDP

Unlike language reasoning under tool invocations, where tools are optionally provided and agents autonomously decide whether to use them, this kind of tasks inherently requires tool invocation. For example, an agent may need to clean a room using APIs designed for a robotic arm or send an email using an email API.

- **Actions**: An action is formalized using predicates and arguments, e.g., *Pick_Up[Apple]*. Since predicates can have various definitions, the action space is both discrete and heterogeneous.

- **States** and **Transitions**: The definitions of state and transition remain the same as in language reasoning ($\mathcal{T}$ via concatenation), while the properties of actions are different,. Specifically, the initial state $s_1$ is the given task description, and subsequent states are obtained by concatenating $s_1$ with actions from previous steps, as expressed in Equation 1.

- **Rewards**: $R_{\text{default}}$ is normally used by default.

## 2.7   Code Generation in MDP

This is similar to language reasoning under Deterministic $\mathcal{T}$ via Concatenation. The only difference is that an action is a token in the vocabulary set of the LLM (rather than a thought consisting of several tokens). Such definition is originally proposed by Zhang et al. (2023). Under their definition, "the reward of state $s$ is the pass rate of the program on the public test cases. The reward of a partial program is always 0."

## 2.8   Goal-Oriented Dialog in MDP

Previous work (Wang et al., 2020) frames goal-oriented dialog as MDP. Yu et al. (2023) begin using such formulation for LLM-integrated search. The formulation is demonstrated below.

- **Actions**: An action $a \in A$ indicates the intent, which is predefined. For example, the intent to convince the Persuadee using reasoning and factual evidence is defined as "Logical Appeal". This is commonly termed "dialog act" (Wang et al., 2020).

- **States**: $s_t$ is defined as the dialogue history of previous $t$ turns, containing dialog act and agent/user utterances

$$h = \left( a_0^{\text{agent}}, u_1^{\text{agent}}, u_1^{\text{usr}}, \ldots, a_{t-1}^{\text{agent}}, u_t^{\text{agent}}, u_t^{\text{usr}} \right) \tag{3}$$

, where $u_i^{\text{agent}}, u_i^{\text{usr}}$ are the utterances of the agent and user, respectively at the $i-$th turn.

- **Transitions**: The state is updated according to stochastic responses from the user $u_i^{\text{usr}}$ (Wang et al., 2020)

- **Rewards**: it represents the immediate feedback of a desired conversational outcome, such as in-process, success, or failure of persuading a user to donate to a charity

## 2.9 Discussion

**Accessibility of Action-State Transition**    In environments under deterministic transitions (e.g., dialog, code generation), the next state $s'$ can be directly derived based on the selected action. In a dynamic environment, $s'$ can be either sampled over the probability distribution or generated from $\text{lmpr}_{\text{transition}}$. Section 4 will demonstrate how this property affects search procedures.

**Overhead of Using MDP Definition**    Although the comprehensive definition provides a unified interface to discuss LIS frameworks, it increases the overhead when applied to graph traversal tasks, since several defining characteristics of an MDP are not necessary, e.g., state transitions and explicit definitions of actions.

**Why Is There No Previous Work Defining Reasoning Tasks as MDPs?**    Typical MDPs are often defined for decision-making models which can only handle the tasks whose action space is constrained and finite. However, general-purpose models like LLMs naturally deal with infinite or/and hard-to-define action space, since LLMs can infer plausible actions with world knowledge and commonsense.

**Do Tasks Enable Action Undoing and State Back-Up?**    Some environments allow going back up to an earlier step after executing a sequence of actions (e.g., reasoning tasks), while other tasks may not (such as robotic tasks). This property is particularly important to discuss the applicability of LLM-integrated search methods. For environments under such property, the LIS agent can feel free to simulate future states for planning without worrying that the change of environments is irreversible. Details will be discussed in § 5).

## 3    LLM-Profiled Roles

Following standard reinforcement learning terminology (Sutton & Barto, 2018), an agent designed to solve Markov Decision Processes (MDPs) typically incorporates the following components:

- **Policy** $\pi(a_t \mid g, s_t)$: Determines the action $a_t$ to take given the current state $s_t$.

- **Value Function** $V^\pi(s) \mapsto R$: Estimates the expected return of state $s$ under policy $\pi$.

- **Transition Model** $\mathcal{T}(s_{t+1} \mid s_t, a_t)$: Represents the dynamics of the environment, predicting the next state $s_{t+1}$ given the current state $s_t$ and action $a_t$.

These definitions are broadly applicable across different agent designs. In this work, we adapt them to LLM-based search and focus on how to profile LLMs to work as/for these agentic components.

**Background of LLM-Profiled Policy, Evaluator and Transition Model**  This section outlines the implementation of the three core components using three types of LMPRs. These roles are defined by Li (2024) as the LLM-profiled policy ($\text{lmpr}_{\text{policy}}$), evaluator ($\text{lmpr}_{\text{eval}}$), and transition model ($\text{lmpr}_{\text{transition}}$). For brevity, these notations are commonly adopted throughout this work.

While prior studies such as Spiegel et al. (2024) and Feng et al. (2024) explored these LMPRs primarily in theoretical contexts and toy environments for reinforcement learning, this section extends these ideas by presenting detailed implementations in real-world tasks.

**Presentation of Prompting Examples**  To illustrate how LLMs are configured for different LMPR roles, we provide prompting examples throughout the paper. Model outputs are visually distinguished using shadow boxes for clarity. For example:

> An output from LMPR

To maintain brevity, placeholders enclosed in angle brackets (e.g., `<demos>` for few-shot demonstrations and `<task desc.>` for task descriptions) are used to represent verbal components within prompts.

Table 3: Types of LLM-profiled policy. Note that what we really require is only $a_t$. $N$: sample size; $T$: length of action sequence.

| Type | Outputs | Example Works |
|---|---|---|
| Deterministic | $a_t$ | Xie et al. (2023), ReAct (Yao et al., 2023b) |
| Batch | $a_t^1, a_t^2, \ldots, a_t^N$ | Yao et al. (2023a), Jacob et al. (2024) |
| Stochastic | Probability Distribution over all the actions $\in A_{\text{candidate}}$ | MC-DML (Shi et al., 2025) |

## 3.1  LLM-Profiled Policy (LMPP)

This section introduces three types of LMPPs. A deterministic policy maps each state directly to a specific action, while a stochastic policy assigns a probability distribution over actions $\in A_{\text{candidate}}$, where $A_{\text{candidate}}$ may be fixed or change based on the current state $s_t$. Lastly, a batch policy produces multiple candidate actions at each time step. While various types of information can be provided in the LLM prompt, the verbalized observation $o_t$ is always mandatory.

**Deterministic LMPP**  This paragraph introduces three types of deterministic LMPPs:

- **Naive LMPP ($\text{lmpp}_{\text{naive}}$)**: An LLM is directly prompted to generate the next action $a_t$.

- **Reasoning LMPP ($\text{lmpp}_{\text{reasoning}}$)**: To generate $a_t$, the LLM first produces a complete reasoning path that explains or justifies the generation of $a_t$. The reasoning path serves as an explicit intermediate step, enhancing interpretability and illuminating the decision-making process for $a_t$. Finally, $a_t$ is extracted and parsed.

- **ReAct LMPP ($\text{lmpp}_{\text{react}}$)**: In contrast to $\text{lmpp}_{\text{reasoning}}$, $\text{lmpp}_{\text{react}}$ separates the reasoning step and the action-generation step into distinct LLM inference passes. Each pass corresponds to an uninterrupted generation session. The reasoning text may include a planning path (e.g., $\tilde{a}_{t+1}, \ldots, \tilde{a}_T$), but only $\tilde{a}_t$ is used for search. Another distinguishing feature is the more autonomous behavior of this policy, which does not strictly adhere to a fixed reasoning-then-acting sequence. Instead, it can dynamically alternate between reasoning and acting steps, such as reasoning-acting-acting. For example:

Your task is to: put a cool tomato in microwave.
>

> think: To solve the task, I need to find a tomato, then cool it with the fridge, and finally put it in the microwave. *<more thoughts>*

OK.

> > go to countertop 1

*<observation>*

> > go to countertop 2

*Prompting Example 1*

The term "react" is attributed to the work of ReAct (Yao et al., 2023b). However, in their formulation, each thought is not explicitly treated as an action; instead, only tool invocations are considered actions in reasoning tasks. This distinction highlights the broader applicability of $\text{lmpp}_{\text{react}}$ in our definition.

**Stochastic LMPP ($\text{lmpp}^s$)**

- **Logit-based LMPP ($\text{lmpp}^s_{\text{logit}}$)**: Leveraging model logits to capture uncertainty is a long-established technique in machine learning (e.g., logistic regression). In language models, the logits over the vocabulary can be converted into a probability distribution over a set of verbalized actions.

- **Verbalized LMPP ($\text{lmpp}^s_{\text{verbalized}}$)**: As shown by Lin et al. (2022), LLMs can "learn to express uncertainty about its own answers in natural language," enabling the probability distribution to be directly generated through prompting.

- **Self-consistency ($\text{lmpp}^s_{\text{sc}}$)**: This approach involves sampling multiple trajectories from an LLM and computing self-consistency scores (Wang et al., 2023) by counting the frequency of each outcome, which is then normalized to form a probability distribution.

**Batch LMPP ($\text{lmpp}_{\text{batch}}$)**  A batch LMPP generates multiple actions simultaneously.

1. **$\text{lmpp}_{\text{batch1}}$**: In this variant, one inference generates multiple candidate actions in text form, with candidates separated by a special token for subsequent extraction. As noted by Yao et al. (2023a), this avoids duplication in constrained action spaces, such as selecting a word in a crossword puzzle. By leveraging a global view of the action space, $\text{lmpp}_{\text{batch1}}$ improves efficiency of LLM inference and coherence in tasks.

2. **$\text{lmpp}_{\text{batch2}}$**: This variant leverages a stochastic LMPP to generate a probability distribution across a set of candidates. Actions are then sampled directly from this distribution.

### 3.2  LLM-Profiled Transition Model (LMPT)

Transition models are especially beneficial in dynamic environments, while in deterministic settings, where transitions are predictable or actions easily reversible, their utility is limited. $\text{lmpr}_{\text{transition}}$ predicts outcomes according to LLMs' internal knowledge. The profiling can be categorized as generating: **1) Full state**: The final goal is to return a full state/observation at the current step, as exemplified in Example 2 from Hao et al. (2023).

<profile information>
[STATE 0] I have that, the white block is clear, the cyan block is clear, <more detail>
[ACTION] Pick up the brown block.
[CHANGE]

> The hand was empty and is now holding the brown block, the brown block was on the table and is now in the hand, and the brown block is no longer clear. [STATE 1] I have that, the white block is clear, the cyan block is clear, <more detail>

*Prompting Example 2*

**2) Partial observation**: The partial observation would be further processed to form the full state. One obvious task is reasoning via QAs.

Given a question, please decompose it into sub-questions. For each sub-question, please answer it in a complete sentence, ending with "The answer is". When the original question is answerable, please start the subquestion with "Now we can answer the question:
<few shot demos>
**Question 1:** James writes a 3-page letter to 2 different friends twice a week. How many pages does he write a year?
**Question 1.1:** How many pages does he write every week?

> *Answer 1.1:* James writes a 3-page letter to 2 different friends twice a week, so he writes 3 * 2 * 2 = 12 pages every week. The answer is 12.

*Prompting Example 3*

### 3.3 LLM-Profiled Evaluator (LMPE)

LLMs can serve as flexible evaluators (*LMPEs*) by leveraging their generative and probabilistic capabilities. We propose categorizing these evaluators along three key dimensions:

- **Task Formulation**: Whether the evaluation is a binary classification, multi-choice QA, or a free-form judgment influences how the LLM's output or logits can be interpreted. These tasks are always formulated in LLMs' system-level prompts.

- **State vs. Action vs. Episode Evaluation**: This is analogous to state/action-state value functions in reinforcement learning. Depending on whether the evaluator is assessing a static state $s_t$ or a transition $(s_t, a_t)$, the LLM must parse different context inputs to provide a valid judgment. Another evaluation target is the cross-trial episode (i.e., a sequence of actions leading to a terminal state ) (Shinn et al., 2023; Shi et al., 2025).

- **Output Types**: Unlike lmpp or lmpt, the output of an evaluator lmpe can produce not only text-based outputs (continuous text or discrete labels or free-form text) but also raw logits and self-consistency scores. This flexibility allows for nuanced scoring and confidence measurement that can be mapped to discrete classes or continuous values.

- **Use of Reasoning**: As LMPPs, some reasoning techniques can be generalized to LMPEs.

As summarized in Table 4, various works have shown that LLMs configured under these dimensions can support diverse evaluation goals. The paragraphs below illustrate the four dimensions through concrete prompts and output examples.

Table 4: LLM-profiled evaluators lmpe. The columns of "*V*?" and "*Q*?" indicate whether the LMPE configuration can work as state value function and action value function, respectively.

| | Prompting Tasks | Outputs | V? | Q? | Example Works |
|---|---|---|---|---|---|
| lmpe1 | Binary/Multi-class Classification | Discrete values mapped by LLM generations | ✓ | ✓ | RAP (Hao et al., 2023), ToT (Yao et al., 2023a), Koh et al. (2024), LATS (Zhou et al., 2024a) |
| lmpe2 | Binary/multi-class Classification | Logits of LLM generations | ✓ | ✓ | RAP (Hao et al., 2023), Tree-BeamSearch (Xie et al., 2023) |
| lmpe3 | Multi-choice QA | Choices of top-N actions | ✗ | ✓ | ToT (Yao et al., 2023a), Think-on-Graph (Sun et al., 2024) |
| lmpe4 | Classification (Implicit) | Logits of given continuation | ✗ | ✓ | RAP (Hao et al., 2023) |
| lmpe5 | Multi-choice QA | Logits of given continuation (choices) | ✓ | ✗ | Kadavath et al. (2022), Xie et al. (2023) |
| lmpe6 | Scoring | Generated continuous scores | ✓ | ✗ | Think-on-Graph (Sun et al., 2024) |
| lmpr$_{policy\&eval1}$ | NA | Logits of policy generations | ✗ | ✓ | RAP (Hao et al., 2023), Tree-BeamSearch (Xie et al., 2023) |
| lmpe$_{verbalize}$ | Episode Evaluation | Free-form text | NA | NA | Reflexion (Shinn et al., 2023), MC-DML (Shi et al., 2025) |
| lmpr$_{policy\&eval1}$ | NA | Logits of policy generations | ✗ | ✓ | RAP (Hao et al., 2023), Tree-BeamSearch (Xie et al., 2023) |
| lmpr$_{policy\&eval2}$ | NA | Self-consistency scores of policy generations | ✗ | ✓ | LATS (Zhou et al., 2024a), rStar (Qi et al., 2025), ToolChain* (Zhuang et al., 2024), |
| lmpr$_{policy\&eval3}$ | NA | Consistency of policy-generated steps and a candidate plan | ✓ | ✗ | LLM-A* (Meng et al., 2024) |
| lmpr$_{policy\&eval4}$ | NA | Q-Values based on a strong lmpp | ✗ | ✓ | Q* (Wang et al., 2024a) |
| lmpr$_{transition\&eval1}$ | NA | Logits of lmpt generations | ✓ | ✗ | RAP (Hao et al., 2023) |
| lmpr$_{transition\&eval2}$ | NA | Logits of lmpt generations | ✓ | ✗ | / |

**Task Formulations**  Three types are commonly used for evaluation. **1) Binary/Multi-class Classification**: As shown below, a prompt can explicitly request a binary judgment, e.g., yes/no or failure/success:

<prompt>

> Status: "failure"

*Prompting Example 4*

In some cases, more fine-grained judgments are required with multiple labels. For example, the "status" tag is defined for LMPE to indicate partial successes in Koh et al. (2024):

<prompt>

> Status: "failure"
> On the right track to success: "yes"

*Prompting Example 5*

This can be considered as multi-class classification, where "yes" yields an intermediate class. A more direct multi-class classification is specified in the example below, where the prompt assesses whether a given set of numbers can reach 24:

> Evaluate if given numbers can reach 24 (sure/likely/impossible)
> 10 14
> 10 + 14 = 24
> sure
>
> 1 3 3

> 1 * 3 * 3 = 9
> (1 + 3) * 3 = 12
> 1 3 3 are all too small
> impossible

*Prompting Example 6*

**2) Multi-choice QA**: This is often advantageous when directly scoring an action/state is difficult to compute in contrast to comparing multiple candidates. For example, it is difficult to judge whether a give passage is coherent, while it is easy to judge whether Passage A is more or less coherent than Passage B. Another way is to implicitly compare different solutions via voting through self-consistency scores of lmpp, which belongs to the next formulation type. **3) Scoring**: A continuous score is given for each candidate option. Example 7 demonstrates how to prompt LLMs for scoring in the task of reasoning over a knowledge graph.

> Please rate thee contribution of the relations on a scale from 0 to 1 (the sum of the scores
> of the relations is 1)
> Q: <Query>
> Topic Entity: <Topic Entity>
> Relations: <list of relations>                                    *Prompting Example 7*

**4) No Explicit Evaluator Definition**: Evaluation can be inferred from the lmpp's generative process itself. In such cases, no separate system-level prompt is required for task formulation. Likewise, lmpt can be used for evaluation. This LLM-based evaluation will be detailed from the perspective of output types.

**State vs. State-Action Function**    **1) State-Value Evaluator**: A state-based evaluator accepts $s_t$ as its input to produce a judgment:

$$\text{discrete judge} = \text{lmpe}(s_t) \tag{4}$$

Example 6 is one of the example. **2) State-Action Evaluator**: the evaluator assesses whether taking action $a_t$ is appropriate at the current state $s_t$:

$$\text{discrete judge} = \text{lmpe}(s_t, a_t) \tag{5}$$

This setup is exemplified in Example 10, where the new sub-question ($a_t$)'s usefulness depends on the prior state ($s_t$). Below is another example of BlocksWorld (Slaney & Thiébaux, 2001):

> [**STATE**]
> As initial conditions I have that, the blue block is clear, the orange block is in the hand,

the red block is clear, the hand is holding the orange block, the red block is on top of the yellow block, the blue block is on the table, and the yellow block is on the table. My goal is to have have that the red block is on top of the yellow block and the orange block is on top of the blue block.
[**ACTION**]
stack the orange block on top of the red block
[**EVALUATION**]

> bad

*Prompting Example 8*

The prompting format is adapted from Hao et al. (2023).

**Outputs**   Finally, the outputs of lmpe can take one of several forms, depending on how we wish to interpret or utilize the evaluator's opinion:

1. **Mapping lmpe generation to discrete values**: For instance, "impossible" or "Yes" may be mapped to numeric scores (0 or 1) in Quotes 6 and 10.

2. **Using logits of lmpe**: The probability of generating a specific token (e.g., "impossible") can serve as the confidence score.

3. **Using logits of given continuations**: Rather than having the LLM generate the evaluation tokens, one can provide the exact sequence to be evaluated (e.g., "good" in Quote 8 or "(A) good"). The log probability of each token is then summed to indicate how well (i.e., how confidently) the model "accepts" that evaluation in the given context. A higher cumulative log-likelihood suggests that the LLM finds the provided evaluation more plausible. Moreover, multiple predefined options (e.g., "(A) Correct" vs. "(B) Incorrect") can be separately fed in as continuations. The log probabilities of each option can then be compared as self-evaluation scores (Xie et al., 2023; Kadavath et al., 2022).

4. **Using logits from lmpp or lmpt**: Alternatively, the evaluation can be derived from the policy or transition model's token probabilities. This method can avoid additional inference steps by reusing existing logits. However, the fundamental flaw is that when multiple plausible answers exist, individual logits can be low even if overall confidence in the correct answer set is high (Lin et al., 2022).

5. **Using self-consistency scores based on lmpp or lmpt**: By sampling multiple trajectories (or states) from an LMPP (or LMPT) via self-consistency (Wang et al., 2023), one can gauge confidence via how often a particular action (or state) appears. More frequent outcomes can be assumed more likely (or better). One challenge is how to distinguish different outcomes. For example, Zhuang et al. (2024) fine-tune a natural language inference (NLI) model to distinguish the generated actions. Note that the logits from the LMPP or the self-consistency scores based on the LMPP ultimately converges to the probability distribution of a stochastic LMPP.

6. **Comparing consistency of lmpp-generated steps and candidate states**: The actions/plan generated by actions can be compared to the candidate for evaluation. This is suitable for tasks with limited successor states.

7. **Comparing Consistency of lmpp-Generated Steps and Candidate States**: The actions or plans produced by the policy can be compared against candidate states to assess consistency. This approach is particularly suitable for tasks with a limited number of successor states, e.g., sovling a maze (Meng et al., 2024).

8. **Using a Stronger lmpp as a Proxy Optimal Policy to Approximate Q-Values**: When rewards are only obtained at the terminal state, the Q-value can be approximated by discounting the rewards along the path. This approach assumes that all subsequent actions are optimal, which necessitates the use of a more robust lmpp as a proxy for the optimal policy.

**Use of Reasoning**   Similar to lmpp$_{\text{reasoning}}$, a reasoning process can be required before generating the final judgment. This reasoning process provides a logical justification to augment the evaluation. For example:

\<prompt\>

> Thoughts: \<your thoughts and reasoning process\>
> Status: "failure"

*Prompting Example 9*

### 3.4   Discussion

**Inference Cost of lmpp**   In practice, the overall computational cost follows the pattern

$$\text{lmpp}_{\text{react}} > \text{lmpp}_{\text{reasoning}} > \text{lmpp}_{\text{naive}}.$$

The gap between lmpp$_{\text{reasoning}}$ and lmpp$_{\text{naive}}$ arises from the additional output tokens produced for reasoning. More importantly, when commercial API is used, lmpp$_{\text{react}}$ exhibits an even higher cost because each *separate* reasoning or action-generation pass is effectively stateless with respect to the cached K–V pairs from previous passes, thereby preventing token-reuse optimizations.

**Applicability of lmpp$_{\text{react}}$.**   A central requirement for ReAct-style prompting (lmpp$_{\text{react}}$) is the availability of step-wise observations after each action. This imposes two prevalent scenarios:

1. **Tasks relying on simulators**: When direct interaction with the real environment is impractical (e.g., actions on tasks are irreversible), a simulator can be substituted to generate the observation following each action. For instance, an LLM-based simulator (lmpt) might use commonsense knowledge to model environmental responses (e.g., turning on a water tap in a sealed sink). However, such simulators are unsuitable for tasks involving external or private data—like querying proprietary databases or retrieving up-to-date information—since an LLM's internal knowledge typically cannot replicate these data sources.

2. **Action-reversible tasks.**   Certain problems can be retried or backtracked, allowing the agent to iteratively act, observe, and refine its actions, as discussed in Section 2. In Section 5, for example, LLM-based search frameworks such as LATS (Zhou et al., 2024a) leverage this property when interacting with real environments across multiple search steps to perform monte-carlo simulation.

**Risk of lmpe**   Although lmpe can effectively evaluate state or action quality, two challenges stand out:

1. **Mediocre discrimination abilities**: As shown by Chen et al. (2024b), using logits as dense rewards (e.g., in lmpr$_{\text{policy\&eval}}$ or lmpr$_{\text{transition\&eval}}$) can reveal that many open-source LLMs struggle to reliably distinguish "good" from "bad" examples.[3]

2. **In-Context Reward Hacking (ICRH)**: According to Pan et al. (2024), an LLM evaluator (lmpe) may attempt to "explain away" negative feedback by globally altering its reasoning and actions, potentially violating constraints. For example, to fix an INSUFFICIENTBALANCEERROR, the LLM might suggest unauthorized money transfers from other accounts, thus compromising safety or policy compliance.

**Not All Generation with "Reasoning" is Truly Augmented.**   By design, LLMs generate tokens in an auto-regressive manner, meaning earlier tokens are not influenced by later ungenerated tokens. Hence, although reasoning tokens after actions (or evaluation) can make the model outputs more interpretable, they do not always alter subsequent decisions or evaluations. In Xie et al. (2023), for instance, a chain of thoughts

$$a_t, \; \tilde{a}_{t+1}, \; \ldots, \; \tilde{a}_T$$

---

[3]GPT-4 turbo was the most advanced model at the time of evaluation.

is produced, where $\tilde{a}_{t+1}, \ldots, \tilde{a}_T$ are "unrecorded" actions. Crucially, $a_t$ is unaffected by any future $\tilde{a}$ tokens, making this effectively a naive policy rather than a true reasoning-augmented approach.

Similarly, consider the evaluator in Example 10:

> Given a question and some sub-questions, determine whether the last sub-question is useful to answer the question. Output 'Yes' or 'No', and a reason.
> **Question 1:** Four years ago, Kody was only half as old as Mohamed. If Mohamed is currently twice as 30 years old, how old is Kody?
> **Question 1.1:** How old is Mohamed?
> **Question 1.2:** How old was Mohamed four years ago?
> **New question 1.3:** How old was Kody four years ago?
> Is the new question useful?

> Yes. We need the answer to calculate how old is Kody now.

*Prompting Example 10*

Although the model's output includes a short "reason," that intermediate reasoning does not necessarily *inform* the generation of 'Yes' or 'No'.

Table 5: Examples of combining LLM-Profiled Evaluators with heuristics. $\|s = s_g\|_1$ indicate whether the agent reaches the goal state at $s$.

| Task | Value |
| --- | --- |
| Reasoning-QA in RAP (Hao et al., 2023) | lmpe1 + lmpr$_{\text{policy\&eval1}}$ |
| Game (Blockworld) in RAP (Hao et al., 2023) | lmpe4 + lmpe2 + $\|s = s_g\|_1$ (weights ignored) |
| Graph traversal in LLM-A* (Meng et al., 2024) | Euclidean distance to the next state + llme |
| Reasoning in Xie et al. (2023) | lmpe5 + lmpr$_{\text{policy\&eval1}}$ |
| Reasoning-QA/Code Gen/Web Nav. in LATS (Zhou et al., 2024a) | lmpe + lmpr$_{\text{policy\&eval2}}$ |

## 4 Search Procedures

This section presents the reusable search procedures applied across various frameworks, including both non-LMPR-specific and LMPR-based procedures. Unlike Section 3, which focused on configuring LMPRs, here we demonstrate how these LMPRs are integrated into the operational processes. However, some content may overlap slightly for coherence. Table 6 summarizes the dependencies between these search procedures and the LMPR components.

**Search Nodes: Integrating States, Action, and Rewards** In this section, we shift our focus to search and clarify how the fundamental search "node" is defined with respect to states and actions. Some methods (e.g., ToT (Yao et al., 2023a)) treat a node as a particular state in a search tree, with transitions determined by the actions taken. To ensure generality, we unify states, actions, and even their estimated values (or rewards) in a single node structure (e.g., RAP (Hao et al., 2023)), facilitating partial expansions or multi-step lookahead. To align with object-oriented design, we represent a node as $n$ with attributes $n.\text{action}$, $n.\text{state}$, $n.\text{parent}$, and $n.\text{val}$, representing the action, state, parent node, and value, respectively.

Table 6: Overview of dependencies in search procedures. Sampl.: Sampling; Exp.: Expansion; Eval.: Evaluation; Sel.: Selection; Sim.: Simulation; Backprop.: Backpropagation.

(a) Dependency of first-order procedures on LMPRs.

|  | LMPP | LMPE | LMPT |
|---|---|---|---|
| LMPP Sampl. | ✓ | ✗ | ✗ |
| LMPE+ Eval. | ✗ | ✓ | ✗ |
| LMPT Sim. | ✗ | ✗ | ✓ |
| Multi-Choice LMPE Sel. | ✗ | ✓ | ✗ |
| Single-Step UCT Sel. | ✗ | ✗ | ✗ |
| Single-Step PUCT Sel. | maybe | ✗ | ✗ |
| Exhaustive Action Retrieval | ✗ | ✗ | ✗ |

(b) Dependency of higher-order procedures on LMPR-based, first-order procedures.

|  | LMPP Sampl. | LMPE+ Eval. | LMPT Sim. |
|---|---|---|---|
| Value-Based Sel. | ✗ | maybe | ✗ |
| LMPP Exp. | ✓ | ✗ | maybe |
| Path Sim. | ✓ | maybe | maybe |
| MCTS Sel. | ✗ | ✗ | ✗ |
| MCTS Backprop. | ✗ | ✗ | ✗ |

## 4.1 First-Order Procedures

First-order procedures operate independently, without relying on other procedures. They serve as the foundational components upon which more complex procedures are built, ensuring a modular and scalable framework for LLM-based search operations. The first three are based on LLM-Profiled Policy (LMPP), evaluator (LMPE), and transition model (LMPT), respectively, while others not necessarily depend on LMPRs.

**LMPP Sampling** The sampling procedure involves generating multiple actions (assuming $N$ actions) for a given state $s_t$. Generally, there are two approaches to sampling actions: one based on generating actions sequentially using the single-action policy (lmpp), and another based on generating all actions simultaneously using the batch policy ($\mathrm{lmpp_{batch}}$). These approaches are detailed in Procedures 1 and 2, respectively.

---

**Procedure 1** LMPP Sampling: Sample Actions One at a Time

1: **procedure** Sample_LMPP_One_At_A_Pass$(s_t, N)$
2:    $\mathbf{a}_t \leftarrow \{\}$
3:    **for** $i \in \{1, \ldots, N\}$ **do**
4:        $a_i \sim \mathrm{lmpp}\,(a \mid s_t)$
5:        $\mathbf{a}_t \leftarrow \mathbf{a}_t \cup \{a_i\}$
6:    **end for**
7:    **return** $\mathbf{a}_t$
8: **end procedure**

---

**Procedure 2** LMPP Sampling: Sample All Actions at Once

1: **procedure** Sample_LMPP_All_At_Once$(s_t, N)$
2:    $\mathbf{a}_t \sim \mathrm{lmpp_{batch}}\,(a \mid s_t, N)$
3:    **return** $\mathbf{a}_t$
4: **end procedure**

---

**LMPE+ Evaluation** LMPEs can be used to estimate the value (or reward) of a state. The evaluator's output—whether in textual or numerical form—can then be combined with rule-based heuristics to refine the overall assessment. For instance, Table 5 illustrates how numeric outputs from LMPEs are incorporated into a heuristic that balances both LLM-based scoring and domain-specific constraints. Such integrated approaches are particularly relevant when neither pure heuristic nor pure LLM-based evaluation alone is sufficient for robust decision-making. Based on the input, the procedure can be defined as V_EVAL for taking a state as input and Q_EVAL for taking both a state and an action as input.

**Multi-Choice LMPE Selection** lmpe5 (an LLM profiled for multiple-choice tasks) can be leveraged for top-k selection to directly select K nodes.

---

**Procedure 3** TopK Selection via lmpe5

---

1: **procedure** TOPK_SELECT_LMPE($N, k$)
2:     $A \leftarrow \{n.\text{action} \mid n \in N\}$                    ▷ Extract actions from each node in $N$
3:     $A^* \leftarrow \text{lmpe5}(A, k)$                    ▷ Select the top $k$ actions from $A$
4:     $N^* \leftarrow \{n \in N \mid n.\text{action} \in A^*\}$        ▷ Return nodes corresponding to the selected actions
5:     **return** $N^*$
6: **end procedure**

---

**Single-Step UCT Selection** The objective of Upper Confidence Tree (UCT) selection is to choose an action that balances exploitation and exploration. This is captured by the UCT formula:

$$\text{UCT}(s, a) = Q(s, a) + c\sqrt{\frac{\ln N(s)}{N(c(s, a))}}, \tag{6}$$

where $c$ is a constant controlling exploration, $N(s)$ is the number of times state $s$ has been visited, and $N(s, a)$ is the number of times action $a$ has been selected under $s$. A related variant, Predictor Upper Confidence Tree (PUCT), incorporates the prior probability $P(s, a)$ into the exploration term to further guide the action selection. $P(s, a)$ can be implemented by a stochastic LMPP.

$$\text{PUCT}(s, a) = Q(s, a) + cP(s, a)\frac{\sqrt{N(s)}}{1 + N(s, a)} \tag{7}$$

$P(s, a)$ is normally a domain-specific predictor. Based on the two estimates, the procedure is just to iterate over the given actions $A$, along with their values $Q$, and extract the one with the highest value, as summarized in Procedure 4.

---

**Procedure 4** Single-Step UCT Selection

---

1: **procedure** UCT_SELECT($s, A$)
2:     $a^* = \arg\max_{a \in A} [\text{UCT}(s, a)]$
3: **end procedure**
4: **procedure** PUCT_SELECT($s, A$)
5:     $a^* = \arg\max_{a \in A} [\text{PUCT}(s, a)]$
6: **end procedure**

---

**LMPT Simulation** This procedure is straightforward: given the current state $s_t$ and an action $a_t$, the LMPT directly outputs the next state $s_{t+1}$. Unlike LMPP sampling (which may loop through multiple actions) or LMPE+ Evaluation (which may incorporate additional heuristics), no further processing or components are involved.

**Exhaustive Action Retrieval** When the action space is small and well-defined (e.g., in BlockWorld), all possible actions can be retrieved exhaustively. This procedure is primarily used to facilitate the subsequent expansion or simulation steps.

### 4.2 Higher-Order Procedures

**Value-Based Selection**   The first type is top-k selection. The top $k$ states or actions are picked from a large pool of candidates based on their estimated values. A state-value function $V(s')$ or an action-value function $Q(s, a)$ is used to generate values. Commonly, they are implemented by LMPE+ evaluation. The detail is illustrated in Procedures 5.

---

**Procedure 5** Value-Based TopK Selection

---

1: **procedure** $\text{TOPK\_SELECT}(N, k, \text{ValueFunc} = \text{NA})$
2:     **for** each node $n' \in N'$ **do**
3:         **if** ValueFunc $\neq$ NA and $n'$.val **is uninitialized then**
4:             $n'$.val $\leftarrow$ ValueFunc($n'$.state)
5:         **end if**
6:     **end for**
7:     $N^* \leftarrow \arg\text{top}_k\{n'.\text{val} \mid n' \in N'\}$
8:     **return** $N^*$
9: **end procedure**

---

Once $k = 1$, the $\arg\text{top}_k\{n'.\text{val} \mid n' \in N'\}$ reduces to $\arg\max_{n' \in N'} n'.\text{val}$. Note that the **if** statement for value assignment also allows specialized ways to assign values without necessarily relying on $n'$.state. This is particularly prepared for Monte-Carlo Tree Search (MCTS), where Q values are accumulated during backpropagation.

Another type is threshold-based selection. Here, the procedure repeatedly samples one action from the current node's state, simulates the new state, and evaluates its value. If the value surpasses a threshold $\theta$, the procedure returns the newly created node; otherwise, it continues sampling.

---

**Procedure 6** Value-Based Threshold Selection

---

1: **procedure** $\text{THRESHOLDSELECT}(n, \theta)$
2:     **while** true **do**
3:         $a \leftarrow \text{Sample\_Action}(n.\text{state})$
4:         $s' \leftarrow \text{Simulate}(n.\text{state}, a)$
5:         $v \leftarrow \text{ValueFunc}(s')$
6:         **if** $v \geq \theta$ **then**                ▷ Value exceeds threshold; accept this node
7:             Instantiate a new node $n'$
8:             $n'$.state $\leftarrow s'$
9:             $n'$.action $\leftarrow a$
10:            $n'$.val $\leftarrow v$
11:             **return** $n'$
12:         **end if**                  ▷ Otherwise, continue sampling another action
13:     **end while**
14: **end procedure**

---

**LMPP Expansion**   This expansion procedure adds one or more child nodes under the given leaf node(s), typically visualized at the next depth level in a tree search. The procedure is based on the following components:

1. **LMPP Sampling**: This step provides the $n$.action attribute for each node, while leaving $n$.state and $n$.val uninitialized.

2. **Simulating States Based on Sampled Actions**: To enable further expansion, the node's state ($n$.state) must be generated. This can be accomplished via simulators (e.g., using LMPT simulation), direct action execution, or even hard-coded methods (e.g., simply concatenating actions).

3. **(Optional) Expanding Multiple Nodes Simultaneously**: In many search strategies (such as beam search with a beam size greater than 1 or breadth-first search), multiple leaf nodes are expanded concurrently.

A general form of the expansion procedure is specified in Procedure 7.

---

**Procedure 7** Expansion

---

1: **procedure** EXPAND__LMPP($N_t$)
2:     $N_{t+1} \leftarrow \{\}$
3:     **for** each node $n_t \in N_t$ **do**
4:         $A_t \leftarrow$ SAMPLE__LMPP($n_t$.state, $k$)
5:         **for** each action $a_t \in A_t$ **do**
6:             Instantiate a new node $n_{t+1}$ with $n_{t+1}$.action $\leftarrow a_t$
7:             $n_{t+1}$.state $\leftarrow$ SIMULATE($n_t$.state, $a_t$)
8:             $N_{t+1} \leftarrow N_{t+1} \cup n$
9:         **end for**
10:    **end for**
11:    **return** $N_{t+1}$
12: **end procedure**

---

Here, SAMPLE__LMPP refers to Procedure 1 or Procedure 2. The total number of $N_{t+1}$ equals the product of the number of nodes in $N_t$ and the number of actions $k$ sampled per node.

**Path Simulation**

- **Sampling**: At each step, actions $A_{\text{candidate}}$ are sampled from the current state using either LMPP sampling or a random sampling method or a uniform sampling (Shi et al., 2025). We denote by $M$ the size of $A_{\text{candidate}}$ in this paragraph. After sampling, there are different scenarios: In the standard MCTS framework (Sutton & Barto, 2018), a basic policy is used to select an action at random during each simulation step. Additionally, a LMPP can be employed to roll out a path until either a goal state or a terminal state is encountered. We refer to these two approaches as **Greedy-Random** and **Greedy-LMPP**, respectively, noting that more sophisticated implementations exist. **Evaluate-Select-Transit (EST)** or **Transit-Evaluate-Select (TES)**. The former requires $M$ evaluations and 1 transition, while the latter requires $M$ transitions and $M$ evaluations.

- **Evaluate-Select-Transit (EST)**: **1) Evaluate**: A reward model (RM) is instantiated by LMPE+ evaluation to evaluate $M$ pairs $(s_t, a)$ where $a \in A_{\text{candidate}}$. The generated reward is denoted as $r$. Afterwards, a function named `createNode` initializes $M$ nodes with each $s_t$, $a$, and $r$ for node attributes: .state, .action, amd .val.

  **2) Select**: TOPK__SELECT is then applied with only the highest-valued node preserved (i.e., $k = 1$).

  **3) Transit**: Finally, the next state $s_{t+1}$ is generated by a Transition Model (TM). This model can be implemented in one of three ways: through LMPT simulation, by using domain-specific simulators, or by directly executing the selected action in the environment to obtain the following state (or observation). For brevity, we refer to these implementations as TM-LMPT, TM-Sim, and TM-Env, respectively.

- **Transit-Evaluate-Select (TES)**: **1) Transit**: The Transition Model (TM) is applied to $a \in A_{\text{candidate}}$.

  **2) Evaluate**: The reward model (RM) is instantiated to evaluate each $s_{t+1}$, generating reward $r$. The function `createNode` can be applied with $s_t$, $s_{t+1}$, $a$, and $r$ corresponding to node attributes: .state, .parent, .action, .val.

  **3) Select**: TOPK__SELECT is then applied with only the highest-valued node preserved (i.e., $k = 1$).

Procedure 8 demonstrates a general path simulation process.

---

**Procedure 8** Path Simulation

---

1: **procedure** SIMULATE($n_t$, RM, heuristic = NA)
2:     path ← {}
3:     **while** $n_t$.state is not terminal **do**
4:         $s_t = n_t$.state
5:         $A \leftarrow$ SAMPLE_ACTION($n_t$.state)
6:         **if** heuristic == NA **then**                                              ▷ Random action sampling (M=1)
7:             $s_{t+1}^* \leftarrow$ TM($s_t, A$)
8:         **end if**
9:         **if** heuristic == eval_first **then**                                    ▷ Evaluate-Select-Transit
10:             $N_{\text{new}} \leftarrow \{\text{createNode}(s_t, a, \text{RM}(s_t, a)) \mid a \in A_{\text{candidate}}\}$
11:             $n^* \leftarrow$ TOPK_SELECT($N_{\text{new}}, k = 1$)
12:             $s_{t+1}^* \leftarrow$ TM($s_t, n^*$.action))
13:         **end if**
14:         **if** heuristic == transit_first **then**                               ▷ Transit-Evaluate-Select
15:             $N_{\text{new}} \leftarrow \{\text{createNode}(s_t, a, \text{RM}(s_t, a), \text{TM}(s_t, a)) \mid a \in A_{\text{candidate}}\}$
16:             $n^* \leftarrow$ TOPK_SELECT($N_{\text{new}}, k = 1$)
17:         **end if**
18:         path ← path $\cup s_{t+1}^*$
19:         $n_t$.state $= s_{t+1}^*$
20:     **end while**
21:     **return** path
22: **end procedure**

---

**MCTS Selection**  However, during MCTS for planning, before going to the expansion phase, Procedure 4 (One-step UCT Selection) should be used multiple times to traverse from $s_0$ to a leaf node $s_{\text{leaf}}$.

**MCTS Backpropagation - Value Update**  After each simulation returns a reward $r$, update the Q value as:

$$Q_{\text{new}} = \frac{r + Q_{\text{old}} \cdot \text{Count}_{\text{new}}}{\text{Count}_{\text{new}}}, \tag{8}$$

- $r$, depending on the task, can be a reward at the terminal state. In some cases, it can be an aggregated one, if each simulation step yields a reward. Specifically, if rewards $r_t$ are discounted by $\gamma$, then the final sample reward $r$ for backpropagation is:

$$r = G = \sum_{t=0}^{T-1} \gamma^t r_t.$$

  In many implementations (e.g., RAP (Hao et al., 2023)), the step-wise rewards are obtained via LMPE+ or a heuristic evaluation during path simulation. These rewards from simulated nodes are then employed to update the Q values for non-simulated nodes, including the leaf nodes and those above.

- $Q_{\text{old}}$ are the previous $Q$-value.

- $\text{Count}_{\text{new}}$ is the total visit count after the current update.

During some implementations (Hao et al., 2023), each reward $r \in R_{\text{cum}}$ propagating from the terminal node can be stored in a list $R_{\text{cum}}$, and average them when used. The procedure of value estimate, as summarized in Table 5, provides the initialized values for new actions or resulting states.

**MCTS Backpropagation - Visit Update**  Except for the estimated value $Q(s, a)$, the backpropagation also updates the visit count for every state on the path from the root to the leaf and each edge $(s, a)$ along that path, denoted as Count $(s_t)$ and Count $(s_t, a_{t+1})$, respectively. The increase in Count$(s, a)$ (the action-level visit count) reduces the exploration bonus for $a$ in future selections, thus making ( $\mathbf{s}, \mathbf{a}$ ) slightly less

likely to be chosen purely for exploration next time, assuming the same or lower estimated value. Assuming $\text{Count}(s_t, a_{t+1}) -> \text{Count}(s_{t+1})$ in some deterministic environments, only $\text{Count}(s_{t+1})$ is tracked. For the same reason, $Q$ values can be attached as an attribute of the state node. An example is the implementation of RAP (Hao et al., 2023) [4].

Table 7: Search-based frameworks for LLM inference. We use **"*"** to indicate workshop publication. **EAR** means Exhaustive Action Retrieval. **Sel.** indicates selection methods used to select from multiple candidate actions/nodes to create branches in a tree, which can be **UCT** or **PUCT**, **Multi-choice LMPE**, Value-based TopK (**V-TopK**), and **threshold** selection. Many MCTS-specific procedures are excluded, except for path simulation (**Sim.**) to distinguish three types of Transition Models (**TMs**) and processes (**Proc.**).

| | Resemble | Expansion | | LMPE+ Eval. | Sel. | Sim. | | Published Date |
|---|---|---|---|---|---|---|---|---|
| | | EAR | LMPP Exp. | | | TM | Proc. | |
| Beam-LLM (Xie et al., 2023) | Beam Search | ✗ | ✓ | ✓ | V-TopK | ✗ | ✗ | May 2023 (NIPS2023) |
| PathFinder (Golovneva et al., 2023) | Beam Search | ✗ | ✓ | ✗ | V-TopK | ✗ | ✗ | Dec 2023 (NeurIPS2023*) |
| Think-on-Graph (Sun et al., 2024) | Beam Search | ✓ | ✗ | ✓ | Multi-Choice LMPE | ✗ | ✗ | Jul 2023 (ICLR2024) |
| Think-on-Graph 2.0 (Ma et al., 2025) | Beam Search | ✓ | ✗ | ✓ | Multi-Choice LMPE | ✗ | ✗ | Jul 2024 (ICLR2025) |
| ToT (Yao et al., 2023a) | BFS (B for Breath); DFS | ✗ | ✓ | ✓ | V-TopK for BFS; Threshold for DFS | ✗ | ✗ | May 2023 (NIPS2023) |
| Best-LLM Koh et al. (2024) | BFS (B for Best) [1] | ✗ | ✓ | ✓ | V-TopK | ✗ | ✗ | Jul 2024 |
| LLM-A* (Meng et al., 2024) | A* | ✓ | ✗ | ✓ | V-TopK | ✗ | ✗ | Jun 2024 (EMNLP2024) |
| Q* (Wang et al., 2024a) | A* | ✗ | ✓ | optional | V-TopK | ✗ | ✗ | Jun 2024 |
| ToolChain (Zhuang et al., 2024) | A* | ✗ | ✓ | ✓ | V-TopK | ✗ | ✗ | Oct 2023 (ICLR2024) |
| RAP (Hao et al., 2023) | MCTS | ✓ | ✓ | ✓ | UCT | LMPT | EST | May 2023 (EMNLP2023) |
| LATS (Zhou et al., 2024a) | MCTS | ✓ | ✓ | ✓ | UCT | Env | TES | Oct 2023 (ICML2024) |
| LLM-MCTS (Zhao et al., 2023) | MCTS | ✓ | ✓ | ✓ | PUCT | Simulator | Greedy | May 2023 (NIPS2023) |
| rStar (Qi et al., 2025) | MCTS | ✗ | ✓ | ✓ | UCT | LMPT | Greedy | Aug 2024 (ICLR2025) |
| MC-DML (Shi et al., 2025) | MCTS | ✗ | See para. 5.6 | ✓ | PUCT | Simulator /Env | Greedy | |
| PG-TD (Zhang et al., 2023) | α-MCTS | ✗ | ✓ | ✓ | PUCT | ✗ | ✗ | Mar 2023 (ICLR2023) |
| GDP-ZERO (Yu et al., 2023) | α-MCTS | ✓ | ✓ | ✓ | PUCT | ✗ | ✗ | Oct 2023 (EMNLP2023) |
| TS-LLM (Wan et al., 2024) | α-MCTS | ✓ | ✓ | ✓ | UCT | ✗ | ✗ | Sep 2023 (ICML2024) |
| ReST-MCTS* (Zhang et al., 2024a) | α-MCTS | ✓ | ✓ | ✓ | UCT | ✗ | ✗ | Jun 2024 (NIPS2024) |

---

[4]`https://github.com/maitrix-org/llm-reasoners/blob/main/reasoners/algorithm/mcts.py`
[1]multimodal
[2]fine-tuning

# 5 Frameworks Based on Search Algorithms

This section summarizes how different frameworks utilize search algorithms, leveraging the LMPRs and search procedures introduced in Table 7. Note that, some MCTS-specific procedures (e.g., **MCTS selection**, and **MCTS backpropagation**) are not elaborated in the table. UCT selection is highlighted to distinguish between PUCT and UCT variants; **simulation** is included to clarify whether LMPT or environment-based simulation or simulator is used. Below, we discuss perspectives that are not fully captured in Table 7.

## 5.1 Beam Search

Beam-LLM (Xie et al., 2023) and PathFinder (Golovneva et al., 2023) adapt beam search for reasoning via concatenation, while Think-on-Graph (Sun et al., 2024) and Think-on-Graph 2.0 (Ma et al., 2025) are applied for reasoning over knowledge graph. Generally, **Expansion** and **TopK-Based Selection** always alternate iteratively until reaching a terminal state.

**Beam-LLM (Xie et al., 2023)**

- **LMPP Expansion**: Each set of beam nodes $N_t$ is passed to the LMPP expansion procedure. Internally, SAMPLE_LMPP calls either SAMPLE_ACTIONS_ONE_AT_A_PASS or SAMPLE_ACTIONS_BATCH, while SIMULATE can be a simple concatenation transition: each node $n_t \in N_t$ has its parent state $n_t$.parent expressed as a sequence of actions $(a_1, \ldots, a_{t-1})$, an action $a_t$ assigned to $n_t$.action, and the new node's state ($n_t$.state) as $(a_1, \ldots, a_{t-1}, a_t)$. The resulting set of expanded nodes is denoted $N_t^{\mathrm{sample}}$.

- **TopK-Based Selection**: From $N_t^{\mathrm{sample}}$, a value-based selection procedure is applied to pick the top-$k$ nodes (the beam size). This subset is returned as $N_t$. Specifically, it uses a value function implemented by $\mathrm{lmpe5} + \mathrm{lmpr}_{\mathrm{policy\&eval1}}$.

**PathFinder (Golovneva et al., 2023)** This framework also applies **LMPP Expansion** and **Value-Based TopK Selection** iteratively. However, it computes a summed similarity score as the value for each candidate node, comparing its state with those of other beams $N_t^{\mathrm{sample}}$. The similarity function can be as simple as n-gram overlap.

**Think-on-Graph (Sun et al., 2024)** In the graph traversal tasks described in Section 2, the search space is derived directly from explicit task specifications. In contrast, when searching over a knowledge graph, the search space is constructed from the graph itself.

- **Step 1: Entity Initialization**: An LLM is prompted twice to extract the topic entities in the question, and select the top-K entities (Procedure 3), where $K$ is the beam size. The resulting entities $\mathcal{E}_0$ form the initial paths.

- **Step 2: Expansion via SPARQL Queries for Candidate Relations**: Unlike previous two frameworks, LMPP expansion and sampling are not necessary because neighboring nodes $N_t^{\mathrm{candidate}}$ can be identified via simple SPARQL queries, along with candidate relations, i.e., Exhaustive Action Retrieval is applied. The relation set can be denoted as $\mathcal{R}_{\mathrm{entity}}$, where topic $\in \mathcal{E}_{t-1} = \{n_{t-1}.\mathrm{state.tail\_entity}|n \in N_{t-1}\}$. The queries are run for each tail entity $e \in \mathcal{E}_{t-1}$ to get candidate relations. The size of $\mathcal{E}_{t-1}$ equals to beam size $B$.

- **Step 3: TopK Selection from Candidate Relations**: Procedure 3 is applied to directly select the top K relations from candidates for each entity $e \in \mathcal{E}_{t-1}$. Note that both $K$ here is not necessarily the beam size $B$ or $B$ divided by $K$, since the final path will be formed after the following entity expansion and selection phases.

- **Step 4: Expansion via SPARQL Queries for Candidate Entities**: Candidate entities are selected via SPARQL queries for each selected relations in the last step.

- **Step 5: TopK Selection from Candidate Entities**: To finish each triple tailed by the K relations selected above.lmpe6 would be used to score each triple regarding their contributions to solve the given question. This defines ValueFunc in Procedure 5. Note that the $k$ here is the beam size.

- **Step 6: LLM Reasoning for Terminal States**: At the end of each iteration, the LLM is prompted to judge whether a terminal state is reached, where the knowledge is enough to reach the final answer.

**Think-on-Graph 2.0 (Ma et al., 2025)**  Think-on-Graph 2.0 (Ma et al., 2025) differs from Think-on-Graph in the following perspectives.

- **Entity Initialization**: The initial entities $\mathcal{E}_0$ are initialized and linked to the knowledge graph via an entity linking method.

- **Context Retrieval**: A dense retrieval model (DRM) is used to retrieve context with $\mathcal{E}_0$ as input.

- **LLM Reasoning for Terminal States**: Same as Step 6 in Think-on-Graph.

- **Relation Identification**: This is basically identical to Step 2-3 in Think-on-Graph. [5] One difference in Step 3 is that Procedure 3 and the LMPE in Step 3 will evaluate the candidate relations of all the entities $\mathcal{E}_{t-1}$ in one LLM inference.

- **Entity Identification**: This corresponds Step 4-5 in Think-on-Graph. Step 4 still locates potential entities on knowledge graph [6]. The main difference is that the the pruning process is based on a context retrieval process over unstructured documents.

- **LLM Reasoning for Terminal States**: Similar as Step 6. The only difference is that the retrieved context for each entity is added for prompting.

## 5.2 Breadth-First Search

**Tree-of-Thoughts (ToT) (Yao et al., 2023a)**

- Similar to beam search, breadth-first search (BFS) is performed by iteratively applying **LMPP Expansion** and **TopK Selection**.

- A key difference is that, in BFS, all nodes at depth $t$ undergo the same number of actions before expanding further levels. This enforces uniform depths across expansions.

## 5.3 Depth-First Search

**Tree-of-Thoughts (ToT) (Yao et al., 2023a)**  Yao et al. (2023a) also apply depth-first search (DFS) for LLM inference, relying on the **LMPP Expansion** and **Threshold-Based Selection**. Key points include:

- **Threshold Selection**: One action (node) is sampled at a time, but it is not compared with other nodes. Instead, once its value is evaluated by whether it surpasses a threshold, that path is followed to its conclusion.

- **LMPE Evaluation for Deadend**: The system uses a deadend judgment to halt exploration of unpromising paths, which can be considered as another LMPE evaluation.

- **Backtracking**: Upon reaching a deadend, the system reverts to an earlier node and continues exploring other previously expanded but untried branches.

- **Path Maintenance**: Because of backtracking, the framework must track partial paths, whereas BFS or beam search only needs to maintain the selected nodes at each depth.

---

[5]For Step2, they do not explicitly specify SPARQL implementations.
[6]Still, they do not explicitly specify SPARQL implementations.

### 5.4 Best-First Search

Best-first search typically uses a heuristic function $h(s)$ to estimate how promising a state will reach the goal.

**Best-LLM (Koh et al., 2024)**

- Uses **LMPP Expansion** for the selected node or the initial root node, where actions are executed in the environment (web interface) to simulate the next states. The generated nodes are saved in $N$.

- Employs TopK_Select (k=1) through LMPE+ evaluation on each node in saved nodes $n \in N$. The node value $n$.val is derived from evaluating its parent's state.

- Continues until either the search tree reaches a specified budget $\beta$ or the state value exceeds a threshold $\theta$.

### 5.5 A*

A* is similar to best-first search but augments the heuristic $h(s)$ with the accumulated cost/utility $g(s)$ to reach a node from the start. The evaluation function is

$$f(s_t) = g(s_t) + \lambda\,h(s_t), \tag{9}$$

where $\lambda$ balances the two terms. Table 8 summarizes how two frameworks implement this formula differently:

Table 8: LMPE+ Evaluation for LLM-A* and Q*. $\mathrm{dist}(\cdot,\cdot)$ represents the actual distance between two points, while $\|\cdot\|$ denotes the Euclidean norm. $s_0, s_t, s_n, s_g, s_{\mathrm{llm}}$ represent the initial, current, neighbor, goal, LMPP-generated states, respectively.

| | $g(s)$ | | $h(s)$ | Use of LMPE |
|---|---|---|---|---|
| | Agg | $R(s)/\mathrm{Cost}(s)$ | | |
| LLM-A* (Meng et al., 2024) | $\sum$ | $\mathrm{dist}(s_0, s_n)$ | $\|s_n - s_g\| + \|s_n - s_{\mathrm{llm}}\|$ | lmpr$_{\mathrm{policy\&eval3}}$ (for $h(s)$) |
| Q* (Wang et al., 2024a) | min , max , $\sum$, Last | Human feedback, ground-truth, rules, LLM logits | $\max_{a_t \in \mathcal{A}} Q^*(s_t, a_t)$ | lmpr$_{\mathrm{policy\&eval4}}$ (for $h(s)$) |
| ToolChain (Zhuang et al., 2024) | $\sum$ | Self-consistency scores, Longest Common Subsequence scores | Consistency scores, relative position scores | lmpr$_{\mathrm{policy\&eval2}}$ (for $g(s)$), lmpr$_{\mathrm{policy\&eval3}}$ (for $h(s)$) |

**LLM-A* (Meng et al., 2024)**

- Designed for path-finding tasks (e.g., mazes).

- $g(s_t)$: Computed incrementally as the path cost from $s_0$ to $s_t$. Formally,

$$g(s_t) = \sum_{i=1}^{t} \mathrm{Cost}(s_i). \tag{10}$$

- $h(s)$ under LMPE+ evaluation: The main modification beyond the typical A* is to integrate a LMPE to the $h(s)$. Specifically, lmpr$_{\mathrm{policy\&eval3}}$ (see Table 4) is applied to evaluate the Euclidean distance from $s_n$ back to the recently visited $s_{\mathrm{llm}} \in$ lmpp, along with the typical Euclidean distance between $s_n$ (expanded neighbour node) and $s_g$.

**Q\* (Wang et al., 2024a)**

- Targets reasoning and code-generation tasks.

- $g(s_t)$: Aggregates rewards via

$$g(s_t) = \text{Agg}\left(\mathcal{R}(s_1), \ldots, \mathcal{R}(s_t)\right), \tag{11}$$

  where $\text{Agg} \in \{\min, \max, \sum, \text{Last}\}$. Under this definition, $\mathcal{R}(s)$ in $g(s_t)$ can be calculated as human feedback, ground-truth, rules or via LMPE+ evaluation, depending on tasks.

- $h(s)$ optionally under LMPE+ evaluation: It is initialized as the optimal Q-value of $s_t$ over all the possible actions:

$$\max_{a_t \in \mathcal{A}} Q^*\left(s_t, a_t\right) \tag{12}$$

  Optionally, $\text{lmpr}_{\text{policy\&eval4}}$ introduces a stronger LMPP to approximate an optimal policy.

**ToolChain (Zhuang et al., 2024)**

- Designed for tool-based tasks but can be generalized to language reasoning tasks ($\mathcal{T}$ via concatenation).

- $g(s_t)$: Aggregated from two terms:

  1. **Self-consistency scores**: $\text{lmpr}_{\text{policy\&eval2}}$ is used to calculate self-consistency scores, normalized by the number of distinct actions.
  2. **Longest Common Subsequence (LCS) scores**: The LCS score between the current generated path $s_t$ and each completed path $m$ in the memory is computed and normalized by the length of the shorter path ($m$ or $s_t$). Among the scores obtained for each memory path, only the maximum score is used for $g(s_t)$.

- $h(s_t)$: Also aggregated from two terms:

  1. **Consistency scores** based on $\text{lmpr}_{\text{policy\&eval3}}$.
  2. **Relative position scores**: Similar to the LCS scores in $g(s_t)$, this metric is also based on memory. It computes the average relative position of a candidate action appearing in memory examples.

### 5.6 Monte Carlo Tree Search

Monte Carlo Tree Search typically involves **MCTS Selection**, expansion (**LMPP Expansion** or **Exhaustive Action Retrieval**), **Path Simulation**, and **MCTS Backpropagation**, including both value and visit update. Most frameworks under review adhere to these steps, except where noted in the highlights that follow.

The details of each framework are specified in Table 7. Note that ReST-MCTS\* (Zhang et al., 2024a) and AlphaZero-Like Tree Search (Wan et al., 2024) feature a search process that is entangled with LLM fine-tuning. In contrast, we decouple the integration of training from the MCTS search, with the training component detailed in Section 7.2.

**Solution Selection**   After MCTS completes its allotted iterations (or reaches its time/resource limit), a tree is left where each child of the root has associated statistics. The standard practice is to pick the move corresponding to the child node with the highest visit count. The reasoning is that more visits generally indicate a move that has been explored more thoroughly and is statistically more promising. In frameworks where final selection strategies are not specified, we assume this default approach.

**RAP (Hao et al., 2023)**

- Applicable to BlocksWorld, Crosswords, and other reasoning tasks.

- **LMPP Expansion** vs. **Exhaustive Action Retrieval**: The method of sampling actions depends on whether the action space is finite and predefined (e.g., BlocksWorld, where exhaustive retrieval is used) or open-ended (e.g., Crosswords, where LMPP sampling is used).

**LATS (Zhou et al., 2024a)**

- Targets tasks with reversible actions (e.g., certain reasoning problems, web navigation).

- **Path Simulation**: Executes actions in the actual environment during path simulation, requiring actions to be reversible to allow repeated trials.

- **Solution Selection**: Besides reaching allotted iterations, the search is terminated when a task is completed successfully. The corresponding path is given as the solution.

**LLM-MCTS (Zhao et al., 2023)**

- Designed for robotic tasks.

- **Path Simulation**: Uses random sampling and a domain-specific simulator for path simulation, producing next states and rewards.

- **MCTS Selection**: Adopts a domain-specific $P(s, a)$ in PUCT, which is derived from LMPP sampling to form an action distribution.

**rStar (Qi et al., 2025)**

- Designed for reasoning tasks. However, a more heterogeneous set of actions is defined, including proposing an intermediate thought, generating multiple thoughts until reaching a terminal state, or rephrasing the original question and questions.

- Acknowledging the limitations of small language models (SLMs) in serving as direct evaluators, it employs self-consistency scores derived from multiple simulation samples (i.e., $\text{lmpr}_{\text{policy\&eval2}}$).

- **Solution Selection**: Instead of selecting only the next action at the root node, the framework selects an entire trajectory from all rollout trajectories. To verify candidates, the selected trajectory is pruned to assess whether another SLM would generate the same subsequent steps.

**MC-DML (Shi et al., 2025)**

- Targets at embodied tasks. Specifically, they experiment on Interactive Friction (IF) games, where reward, state and observation are given by the game environment/simulator.

- **MCTS Selection**: Besides $\text{lmpp}^s_{logit}$, it also uses another two types of $\text{lmpp}^s$ in the case of black-box LLMs, e.g., OpenAI ChatGPT.

- **LMPP Expansion**: This phase is integrated with the selection phase, where only $\text{lmpp}^s$ generates prior distribution. If the action chose by the PUCT selection leads to a leaf node, the environment transits it to the next state, followed by the path simulation phase, where the uniform sampling will be applied.

- **LMPE+ Evaluation**: $\text{lmpe}_{\text{verbalize}}$ is used to generate reflection on history failed episodes during the simulation (i.e., cross-trial information) phrase. We denote the pair of an episode and its reflection as $(\text{eps}_i, r_i)$

- Cross-trial Information for LMPP Generation: Besides the previous steps along the current path (i.e., in-trial information), $(\text{eps}_i, r_i)$ are also added as the prompt of the LMPP. It reminds of cross-trial information within Reflexion (Shinn et al., 2023). However, MC-DML is first used within a LIS framework.

### 5.7 AlphaZero-Like MCTS ($\alpha$-MCTS)

In traditional MCTS, after expanding a node, a path simulation is run to the end of the game to estimate the outcome. However, many modern implementations (like in AlphaGo/AlphaZero) skip the full simulation and instead use a direct evaluation function or Process Reward Model (PRM) to estimate the reward immediately after expansion rather than roll out the entire episode for a reward. This approach can significantly speed up the process and provide more accurate evaluations if the evaluation function is well-tuned. In summary, the following phases are iteratively performed: **Selection**, **Expansion**, **Evaluation** (replacing path simulation), and **Back-propagation**.

**PG-TD (Zhang et al., 2023)**

- Specializes in code generation.

- **MCTS Selection**: Treats the prior distribution $P(a \mid s)$ in PUCT as the LMPP token probabilities for the next token (i.e., $\text{lmpp}^s_{logit}$), given the partial program.

- **Evaluation**: Rather than direct evaluation, the phase consists of the following processes: 1) Uses LMPT simulation to generate the rest of the program, but internally adopts beam search for transformer decoding to complete the program; 2) Generates a reward by running the program on the test cases.

**GDP-ZERO (Yu et al., 2023)**

- Designed for goal-oriented dialogue.

- **LMPP Expansion**: An LMPT is configured for transition, i.e., two inference calls are performed to simulate both the system and user responses. The pairs of responses are stored in the memory $M_{\text{response}}$ for each state. Hence, if the maximum number of responses for state $s$ are reached, the transition is sampled from $M_{\text{response}}(s)$.

- **Evaluation**: LMPE Evaluation is performed. In particular, lmpe2 serves as the PRM, generating a reward signal for MCTS backpropagation after expansion. Its profiling is similar to a simulator, wherein the LLM is configured to emulate a user who is prompted to indicate their willingness to perform a specific action (e.g., making a donation). However, different from the LMPT, a set of five choices is provided for rating, each corresponding to a different reward.

**Tree Search (TS)-LLM (Wan et al., 2024)**

- **Target Tasks**: TS-LLM is designed for both combinatorial tasks (e.g., chess, Game-of-24) and reasoning tasks via concatenation, where each action is treated as a sentence. They also define each action as a token for general language reasoning for Reinforcement Learning from Human Feedback (RLHF).

- **Expansion**: The original work briefly describes two ways to define the search space during expansion: one based on an exhaustive set of possible tokens and the other using candidates sampled by LMPP.

- **Evaluation**: The LLM is fine-tuned as a RM to evaluate intermediate states (see details in Section 7.2).

- **Solution Selection**: The search is performed multiple times to generate candidate solutions. The final solution is chosen according to either the majority voting or the sum of rewards, or the maximum reward, where ORM is used to generate rewards.

- **Solution Selection**: The search process is repeated multiple times to generate candidate solutions. The final solution is selected based on majority voting, the sum of rewards, or the maximum reward, with an ORM used to produce the reward values.

**ReST-MCTS* (Zhang et al., 2024a)**

- **Target Tasks**: ReST-MCTS* is designed for reasoning tasks via concatenation, where each sentence is treated as a discrete step.

- **Evaluation**: Similar to TS-LLM, the LLM is fine-tuned as a PRM to evaluate intermediate states (see Section 7.2 for details). To estimate the value of the current state $s_t$, the rewards from the PRM are weighted by the estimated reasoning distances, analogous to the $h(s_t)$ term in A*.

- **Solution Selection:** After the selection phase, if the value of the selected node meets or exceeds a predefined threshold, the corresponding solution is returned.

## 6 LLM Inference + Search Beyond Sequential Decision Making

The previous focus is on the work that the search process are coupled with the LLM's decoding process, where downstream tasks are formulated as sequential decision-making problems (as detailed in Section 2), and LLMs serve as integral components in search procedures. This section aims to explore all the other works that include both LLM inference and search methods. However, these approaches either optimize other elements (such as model selection or inference workflows) or do not fit in the MDP formulation or use search and LLMs separately. For example, the plans (commonly in the form of PDDL) are prepared by LLMs to perform local search (Valmeekam et al., 2023b; Guan et al., 2023; Valmeekam et al., 2023a).

As a result, the notions of actions and states do not exist, at least during LLM inference. This distinction leads to several fundamental differences. For instance, the roles of LMPP and LMPT are not applicable, which is why LLM sampling, rather than LMPP sampling, is introduced in the frameworks below. Furthermore, state simulation is never required.

**LLM for World Modeling + Search**    In this paradigm, LLMs are employed to generate world models that serve as the basis for planning. In contrast, LLMs in the LIS frameworks operate as world models (e.g., LMPEs and LMPTs). The world models can be represented by the Planning Domain Definition Language (PDDL) (McDermott, 2000), which clearly defines action preconditions and effects, or Python code. For example, Guan et al. (2023); Liu et al. (2023) utilize LLMs to generate PDDL-based world models and then apply classical search-based planners (e.g., LPG) to solve tasks. In contrast, Dainese et al. (2024) prompt LLMs to generate Python code for world modeling and then solve tasks via Monte-Carlo Tree Search (MCTS).

**Meta-Search for LLM-Inference Strategies for Problem Solving**    The frameworks in Section 5 directly search for (partial) solutions—using search itself as the strategy to reach a solution. In contrast, this paragraph focuses on meta-search methods that identify the strategies to reach a solution. For example, DOTS (Yue et al., 2025) investigate whether LMPE evaluation is necessary for a given problem, while the LIS methods integrate LMPE evaluation into determining whether an action is optimal for achieving the goal. Strategist (Light et al., 2025) search for optimal high-level strategies that guide the generation of task solutions. Similarly, AFlow (Zhang et al., 2025) employs MCTS to search for optimal inference workflows for solving downstream tasks. However, the resulting solutions may not strictly conform to a search-based workflow. Essentially, this approach focuses on the structure and efficiency of the overall inference process. In contrast, the LIS frameworks directly concentrate on constructing search workflows for downstream tasks. Besides, Zheng et al. (2025) conduct a grid search over 6 reasoning strategies and 5 types of instructions.

However, their grid search is used solely to evaluate the combinations, rather than as a meta-search strategy for problem solving.

**Equilibrium Search in a Two-Player Game**   Generally, a discriminator (a special LMPE) and a generator (a special LMPP) are defined to engage in a game, where a correctness parameter for the generator is selected uniformly at random. Both the discriminator and the generator receive a payoff of 1 when they agree on the correctness parameter. This equilibrium objective is regularized by the prior, i.e., the inital generation of large language models (LLMs), as some equilibrium solutions may not align with commonsense reasoning. The final state is reached when the discriminator and the generator reconcile with each other. Unlike single-player games, two-player games involve theoretical frameworks under the umbrella of game theory and equilibrium analysis. While this aspect is beyond the scope of this survey, the fundamental steps of LLM sampling and LMPE evaluation remain essential prerequisites before executing equilibrium computations to achieve a regularized equilibrium.

- **LLM Sampling**: An LLM is prompted for sampling in batch. While this is similar to $\mathrm{lmpr}_{\mathrm{batch\_policy2}}$, the LLM is prompted with the value of the correctness parameter, which is either "correct" or "incorrect".

- **LMPE Evaluation**: Corresponding to the values of the correctness parameter, the LMPE (or discriminator) generates binary predictions (i.e., "correct" or "incorrect"). Specifically, the LMPE is implemented as $\mathrm{lmpr}_{\mathrm{eval1}}$.

**Evolutionary Search**   EvoPrompting (Chen et al., 2023) employs evolutionary search to generate implementation code for neural architectures, while Wang et al. (2025b) perform evolutionary search over chemical space, particularly for molecular discovery. Both re-define cross-over and mutation operations via LLMs. However, it is not defined for sequential dicsion making, where the next step is determined by the previous step(s). For example, the complete solution code is not required to be decomposed into smaller components. This traces to the nature of cross-over and mutation operations requring the entire solution as the input.

The key procedures in their algorithm are:

- **LLM Sampling**: An LLM is used during the cross-mutation phase to generate candidate codes. The generation process is conditioned on randomly selected codes from the population.

- **Value-Based Top-K Selection**: This method updates the population by selecting the top $K$ candidates based on a value function. The ValueFunc is implemented by training a deep learning model with the generated code and evaluating its validation accuracy.

**Search over Solution Sketches for Code Generation**   PlanSearch (Wang et al., 2025a) searches over solution sketches prior to generating code—that is, it operates on a natural language description of the correct program. Specifically, the model samples "observations" (small pieces of ideas or hints) which are then concatenated with the task description to prompt the LLM to generate subsequent observations. Each sequence of observations at depth two is then used to prompt the LLM to formulate a comprehensive idea. The LLM is subsequently re-prompted to produce alternative another set of ideas by designating the initial set as incorrect. Finally, each idea is usd to prompt the LLM to generate code.

This framework is discussed in the final section for two reasons: (1) The search space is not directly related to code generation. However, it can be viewed as a form of language reasoning via concatenation, where each thought constitutes an observation; and (2) the framework does not employ traditional search algorithms (e.g., A* or MCTS).

# 7   Related Work

Although the primary focus of this survey is on test-time compute via search, several related directions fall outside our current scope.

## 7.1 Other Frameworks for Test-Time Compute

- **Search with Multi-Modal LLMs.** Some work extends tree-based exploration and action selection to multi-modal contexts by incorporating visual features alongside textual reasoning steps, e.g., Mulberry (Yao et al., 2024)

- **Branching without Search.** Some frameworks utilize branching or tree-like expansions but do not incorporate a full-fledged search algorithm. Examples include Tree-Planner (Hu et al., 2024), Boost-of-Thoughts (Chen et al., 2024a), and Graph-of-Thoughts (GoT) (Besta et al., 2024). Although they adopt branching structures similar to traditional search, these methods rely on aggregation, sorting, or heuristics rather than explicit search procedures.

- **Re-Ranking Frameworks**: In these frameworks, LMPP is initially employed to sample multiple candidate solutions. However, unlike the LMPP sampling described in Section 4, here it generates complete sequences of actions (i.e., plans) instead of just intermediate actions as seen in MDP-like settings. These two variants of LMPP are distinguished as "actors" and "planners" in Li (2024). An evaluation function then is used to re-rank the plans by assigning the scores over plans/final answers. Notable examples include LEVER (Ni et al., 2023) for code generation and DiVeRSe (Li et al., 2023) for language reasoning. It may be a normalized consistency function (Wang et al., 2023) or a learned model like a Process Reward Model (PRM) or Outcome Reward Model (ORM) (Lightman et al., 2024).

- **Sequential Revision**: Same as re-ranking methods, an evaluation function is applied to plans/final answers. However, this process can be iterative. Examples include Self-refine (Madaan et al., 2023a). Table 9 compares re-ranking methods, sequential evaluation and search-based methods for LLM inference. Hence, it can be also defined as a MDP process (Qu et al., 2024).

## 7.2 LLM Fine-tuning for Test-Time Compute

Recent methods adapt LLMs via fine-tuning or preference optimization to enhance their roles in policy, evaluation, and transition modeling. For instance, in the TS-LLM framework (Wan et al., 2024), the LLM is fine-tuned as an Outcome Reward Model (ORM) or a Process Reward Model (PRM) that evaluates each reasoning step. ReST-MCTS* (Zhang et al., 2024a) fine-tunes LLMs to serve both as a policy model, generating reasoning traces, and as a PRM. Although the paper does not explicitly state that both models start from the same checkpoint, the standard procedure implies that a pretrained LLM is duplicated into two copies. In DeepSeek-MCTS (Guo et al., 2025), LLMs are iteratively fine-tuned for both policy and reward roles using question–answer pairs. In this framework, questions are collected, and answers are generated by MCTS guided by a pretrained value model. LLMs are also fine-tuned for other test-time compute tasks. For example, Recursive IntroSpEction (RISE) (Qu et al., 2024) fine-tunes LLMs as the LMPP for sequential revision, while the LMPE can be implemented using a stronger teacher model or via a self-consistency approach (i.e., $\text{lmpr}_{\text{policy\&eval2}}$).

**Fine-Tuning LLMs for PRMs** For all PRMs in the above work, except for DeepSeek-MCTS [7], the unembedding layer, which maps hidden states to a vocabulary distribution, is replaced by an MLP that outputs scalar values. This recalls how reward models are trained in Reinforcement Learning from Human Feedback (RLHF) for general-purpose alignment (Ouyang et al., 2022).

## 8 Discussion

In this section, we analyze how search frameworks for LLM inference deviate from traditional search algorithms, where they apply according to technical constraints, performance and efficiency.

---

[7]Limited details are provided in the paper.

Table 9: Comparisons of LLM inference via search and re-ranking. Here, $A_t$ denotes the set of candidate actions at step $t$ along path $i$, $P$ represents the candidate plans, and $P_{\text{trial}\_j}$ denotes the candidate plans at trial $j$. The term "Iterative?" indicates whether the sampling and evaluation processes are iteratively executed.

|  | Sampling | Evaluation | LMPE? | Iterative? | Fine-tuning |
|---|---|---|---|---|---|
| LIS | actions | $\{a \mid a \in A_t$ along path$_i\}$ | Often | ✓ | TS-LLM (Wan et al., 2024), DeepSeek-MCTS (Guo et al., 2025) |
| Re-Ranking | plans | $\{p \mid p \in P\}$ | Rarely | ✗ |  |
| Sequential Revision | plans | $\{p \mid p \in P_{\text{trial}\_j}\}$ | Always | ✓ | RISE (Qu et al., 2024) |

## 8.1 Deviations from Typical Search Algorithms

**Beyond Finite, Fixed Search Spaces**   Classical BFS or DFS typically requires enumerating all successors at each depth, which can lead to massive memory usage in large or infinite search spaces. By contrast, **LMPP sampling** (based on LLM priors) can manage successor expansions more selectively, reducing the need to store every possibility at each level. Also, it is possible to handle tasks with an open and infinite action space.

**Making "Uninformed" Search Informed**   Traditionally, BFS and DFS are considered *uninformed*, exploring the search space without heuristics. LLM-based frameworks labeled as BFS or DFS often incorporate **LMPP sampling** or **LMPE+ evaluation**, effectively introducing heuristic knowledge from the LLM. Moreover, **anticipating dead ends** in DFS is feasible with LLM-based heuristics. Classic DFS only identifies dead ends when it exhausts neighbor nodes. With an LLM, the search can backtrack early if the model predicts an unpromising or "dead-end" scenario.

**Compromised Optimality in A\***   A* requires an *admissible* heuristic $h(s)$ to guarantee optimality. However, if $h(s)$ partially depends on a policy-generated state $s_{\text{llm}}$ from $\text{lmpr}_{\text{policy}}$, the heuristic may be overestimated. This breaks admissibility assumptions, meaning the final solution may no longer be strictly optimal.

**Terminology Deviation for Heuristic Search: Cost vs. Return/Value**   Many early applications of search algorithms, such as Best-First Search and A*, focused on minimizing quantities like distance, travel time, or energy expenditure. Generally, the cost-based evaluation can be considered as minimization objectives or eliminating "bad" states (or unpromising actions). However, in modern applications involving LLM-integrated search, such as web navigation or document retrieval, heuristics often reflect *value estimates* (positive polarity), especially when LLMEs tend to be defined to reflect the relevance or utility of states, which are better suited for maximization objectives or maintaining "good" states (or promising actions). We highlight this point for rigidity. However, these terms can be abstractly defined without indicating real-world semantics. For example, in game design, moving to a node which end up losing a life point can be given a negative reward, while a reward from gaining a key can be granted as a negative cost.

**RL vs. LMPE-Based Value Functions**   In traditional RL, the value function is rigorously defined via the Bellman equation as the expected cumulative future reward (or cost) that one can obtain starting from a given state. This formulation inherently accounts for the uncertainty of future outcomes by computing a weighted average of rewards over all possible future trajectories (Sutton & Barto, 2018).

In contrast, when LLMs are used in tree search, the "value" assigned to a node is often generated based on the model's world knowledge and reasoning (Yao et al., 2023a). This evaluation serves as a heuristic and is derived from the LLM's internalized representations rather than being computed from a strict, recursive future-reward formulation. As a result, while both the RL value and the LLM-based evaluation score aim to

measure the "goodness" or utility of a state, the latter does not necessarily adhere to the Bellman property (Bellman, 1966; Watkins, 1989).

## 8.2 Applicability

**Extending Uninformed Search to Dynamic Decision-Making**  BFS and DFS were originally designed to explore predefined or easily generated state spaces (e.g., enumerating all children in a graph). This can be: **1) No Need for Transition**: Traditional implementations of DFS and BFS do not require an explicit, computed transition model because they inherently rely on the graph structure where edges already define state transitions. For example, one task is to solve a maze. **2) Only Successor Function**: In some applications, especially in implicit state-space search, a minimal form of transition function (or successor function) is still required to generate successors when the full graph is not explicitly available.

However, when applied to LLM inference, they can be adapted for: **3) Tasks That Require Dynamic Decision-Making**: These tasks are based on state-dependent actions (as seen in planning or reinforcement learning), e.g., solving the Game of 24 and completing crossword puzzles.

**Open-Loop vs. Closed-Loop Frameworks**  This property is important to discuss the applicability of frameworks for planning. **1) Open-Loop (Offline)**: After executing $a_t$, the open-loop agent does not adapt its future actions based on the actual new state. Instead, it continues to follow a pre-planned sequence of actions, which are based on the simulated state during planning. For example, ToT-DFS and ToT-BFS (Yao et al., 2023a) produce a predefined sequence of actions based on a search through a static search tree or graph. This sequence is intended to be executed exactly as planned. Ideally, open-loop planning requires that the environment will remain as anticipated throughout the execution. Such environments satisfies the following assumptions: a) environments are static, b)(no unforeseen events or changes will affect the execution (closed-world assumption), and c) deterministic transitions. **2) Closed-Loop (Online)**: In contrast, after the closed-loop agent executes an action $a_t$, it observes the outcome in the real world (i.e., the resulting state) and can adjust its future action $a_{t+1}$ based on the new state $s_{t+1}$. The plan evolves dynamically as new information becomes available. MCTS-based frameworks, except for LATS (Zhou et al., 2024a), can be open-loop because it generates a sequence of actions based on simulations and a static model of the environment at a given state. However, the agent can operate in a closed-loop manner if MCTS is re-run at each new state: after a plan is generated for $s_t$, only $a_t$ is selected and executed. Instead of executing the rest $a_{t+1}, \ldots$, the agent re-runs MCTS from the new state $s_{t+1}$ to determine the next best action $a_{t+1}$. These frameworks are suitable for task execution in dynamic and interactive environments.

**Many LIS Frameworks Requires Action Undoing**  In many LIS frameworks, the agent has to revisit earlier states, including backtracking to an ancestor node in DFS (Yao et al., 2023a), selecting the next best candidate in Best-first search (Koh et al., 2024), or retracing your steps after an environment-based simulation in MCTS (Zhou et al., 2024a). Each of these processes relies on being able to recover the state reached by previous actions. This limits their applicability to environments where undoing actions is viable, as specified in Table 2.

## 8.3 Performance

Scaling test-time compute via search has generally enhanced the LLMs' power to a new level on reasoning tasks. Nonetheless, there remain certain scenarios where performance is still suboptimal.

**Search Frameworks Perform Worst in Multiple Scenarios**  Empirical studies by Parashar et al. (2025) also indicate that ToT and RAP perform even worse than CoT and self-consistency in various language reasoning tasks. Also, as demonstrated by Snell et al. (2025), if a base LLM can already produce reasonable answers for simple questions, then sequential revision suffices to ensure performance, offering an efficient alternative to more complex re-ranking methods and search methods.

**MCTS May Degenerate in Early Stages**  Chen et al. (2024b) observe that if all candidate steps receive equal (or zero) scores initially, MCTS lacks a clear basis for distinguishing among branches, potentially

leading to suboptimal partial-plan selection. The performance is worse than iterative refinement (Madaan et al., 2023b).

### 8.4 Efficiency

**Compute and Memory Efficiency**  As noted by Chen et al. (2024b), tree search can be 10–20 times slower than iterative refinement, especially if the evaluator (LMPE) has less than 90% discrimination accuracy. High-accuracy LMPEs are essential to prune the search tree effectively, thereby reducing the number of iterations needed. Long-horizon tasks can become particularly expensive when each LLM inference call is executed independently, despite many calls sharing identical prefixes. By leveraging Key-Value (KV) caching, both computational and memory overhead in transformer-based LLMs can be significantly reduced. Yao et al. (2025) propose a tree attention, which leverage tree topology to reduce memory IO with tree-structured KV caches.

**LMPP Expansion and Path Simulation with Memory**  Finally, maintaining a memory of explored nodes can avoid repeated sampling and simulation. If a given state $s$ has already produced certain actions, those child nodes can be cached for subsequent expansions. Similarly, for simulation, previously simulated results can be stored for future simulation. By reusing these cached outcomes, the framework reduces redundant calls to LMPP or LMPT, thereby improving both inference cost.

**Unnecessary LMPP Use in Some Cases**  Some tasks possess a small, tractable action space (e.g., the Game of 24). In such scenarios, *exhaustive* action retrieval may be cheaper than performing multiple LMPP inferences. Designers must weigh the inference costs of LMPP against potential benefits, as LLM-based sampling can be expensive relative to enumerating a finite set of actions. The potential cost of the following LMPE+ evaluation should also be considered.

### 8.5 Future Work

**Parallel Research on LLMs' Competence and Test-Time Computation**  Rather than relying on intensive test-time computation, optimization frameworks—such as Chain of Preference Optimization (Zhang et al., 2024b)—have been proposed to train LLMs to develop tree-like reasoning, thereby reducing the need for real-time deliberation. In parallel with investigating the impact of base models and LMPRs on performance, further research is needed to both enhance LLMs' capacity for autonomous multi-path thinking and identify complex tasks which most benefit from deliberate search during test-time computation.

**Devising Frameworks for Handling Irreversible Actions**  Many prominent LIS frameworks do not incorporate mechanisms for managing irreversible actions. Instead, they circumvent this issue by working in simplified task environments where the effects of actions can be virtually simulated. For example, Tree of Thoughts (ToT) (Yao et al., 2023a) focuses on language reasoning, while Koh et al. (2024) studies web navigation scenarios that involve only reversible actions. This approach is in line with abstract search methods (e.g., DFS or Best-first search on a graph), where action outcomes are computed via deterministic transition functions, eliminating the need to physically execute or reverse actions.

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
