# OpenReview forum: "A Survey on LLM Test-Time Compute via Search: Tasks, LLM Profiling, Search Algorithms, and Relevant Frameworks"
_TMLR — Accepted by TMLR_

### Review · Reviewer_F53r · 2025-01-31

**Summary Of Contributions:**

The paper provides a technical survey/review on LLM inference via search. The paper provides a unified view from task definition, LLM profiling, and search procedures. 11 frameworks are examined with moderate technical details. More specifically, the paper describes a unified MDP task definition and mapped major tasks in the definition. Then the paper discusses LLM-Profiled roles (policy, transition, etc). The structure decouples search procedure and search algorithms.

**Audience:**

Yes

**Broader Impact Concerns:**

This is a survey paper. I don't see concerning claims.

**Claims And Evidence:**

Yes

**Requested Changes:**

The major concern for this kind of work with such a fast moving field is timeliness. From Tables such as Table 7, it is relatively clear that there is a bias towards pre-2024 work. (to be clear, they should be discussed). I would request the authors at least discuss more recent work to the best effort during the revision period, for example, those will appear in ICLR this year.

Table 3 has space issue.

**Strengths And Weaknesses:**

Strength:
1. The reviewer is generally happy with the survey structure and feel it is well organized.
2. The paper is clear to the reviewer and can be a reasonable survey / review paper to the community.

Weakness:
1. The reviewer is a bit surprised to see the number of references as a survey paper for such a topic (that is close to the hot topic on LLM agents), which looks small. There're ~30 references excluding generic surveys / books / paper (e.g., the old CoT paper). Can the authors justify the situation? To be clear, comparing with other papers in Table 1 is not convincing as those papers are not dedicated surveys, with just have one related section.

---

> ### Author Response · Authors · 2025-02-13
> **Response**
>
> Thank you for your constructive feedback and your patience during our revision process. Below is our detailed response. All changes are highlighted in **bold**.
>
> ``Q1: a bit surprised to see the number of references as a survey paper for such a topic (that is close to the hot topic on LLM agents), which looks small ``
>
> We appreciate your observation. While it is true that the literature on “LLM agents” is extensive, our survey deliberately focuses on the more specific topic of LLM inference via search for sequential decision making. This narrower scope naturally results in a more selective set of references that are most directly relevant.
>
> Although we do not consider the very broad scope of LLM agents, to provide broader context, we have further **added Section 7 (Broad scope of LLM inference + search) in the revised manuscript.**
>
>
> ``Q2: Can the authors justify the situation? To be clear, comparing with other papers in Table 1 is not convincing as those papers are not dedicated surveys, with just have one related section.``
>
> Thank you for your feedback. Initially, we included 38 non-survey papers with 11 detailed in Section 5. In the revision, we **expanded the review to 50 papers in total (excluding survey papers), with 20 discussed in detail (Sections 5 & 7).**
>
> * Precise and narrow scope (As demonstrated in the Q1 response): This survey primarily focuses on LLM inference via search, where downstream tasks are formulated as sequential decision-making problems, and LLMs serve as integral components. This focus allows us to present a detailed, concise, and systematic analysis. However, we acknowledge that, given the title of the paper—which does not specify the task scope—our initial content may appear to overstate our scope. Consequently, we have added Section 7 to cover additional research directions that also involve LLM inference and search.
>
> * Only peer-reviewed papers: We have chosen to include only peer-reviewed papers in our survey to ensure reliability.
>
> * Focus Venues: We focus exclusively on high-quality venues that actively publish research on LLM inference via search, including *ACL, ICML, ICLR, and NeuIPS.
>
> * Situation of ICLR 2025: The acceptance notifications arrived after our paper submission. Hence, in our initial version, we only included two ICLR 2025 submissions (peer-reviewed on OpenReview) .
>
> * Missing Works: A systematic search (see Section 5 update) revealed a few additional works that we have now incorporated.
>
> ``Suggestion 1: I would request the authors at least discuss more recent work to the best effort during the revision period, for example, those will appear in ICLR this year. ``
>
> We **added 12 recent papers (9 in detail) based on a systematic search of  ICML/ICLR/NeuIPS papers from 2023–2024 using keywords including “search,” “tree,” and “graph.”**  (We searched for papers from 2023 since the pioneering work, like Tree-of-Thoughts, are published from May 2023.) We also looked online for ICLR 2025 accepted papers; however, since no official list is available yet, we will systematically update this once it is published. Anyway, we **added two pertinent ICLR2025 papers that we found**.
>
> ``Suggestion 2: Table 3 has space issue. ``
>
> Corrected.
>
> We sincerely appreciate the reviewers' valuable insights and warmly welcome any additional guidance that can further enhance our work.

---

> ### Author Response · Authors · 2025-03-05
> **Adding ICLR2025 papers**
>
> We have added 11 additional ICLR2025 papers. In total, the references now include 101 papers, with over 30 highly relevant works on frameworks and/or analysis of LLM inference and search distributed across Sections 5, 6, and 8. For further details, please refer to the "Changes Since Last Submission" on OpenReview.

---

> > ### Comment · Reviewer_F53r · 2025-03-11
> >
> > Thank you! The changes are acknowledged.

---

### Review · Reviewer_G1yV · 2025-03-10

**Summary Of Contributions:**

Summary:

This paper presents a survey on large language model test-time computing scaling with search. The authors call it LLM inference via Search (LIS). They propose to to use a unified formulation based on Markov Decision Processes and it covers a range of tasks. The survey contains three major parts: task definitions, LLM profiling, and modular search procedures. In addition, the paper compares over 25 frameworks and provide details tables and diagrams.

**Audience:**

No

**Claims And Evidence:**

No

**Requested Changes:**

Including more original contributions like experimental validation and new insights.

**Strengths And Weaknesses:**

Strengths:

- The paper researches on a large number of related works and provides an integrated view of the techniques used in LLM compute scaling studies.
- The authors propose a unified framework.
- The paper provides detailed tables and diagrams which is a good summary for readers to get familiar with this field.

Weaknesses:
Although the survey is comprehensive and provides an extensive review of existing literature, merely summarizing and unifying others’ work is not sufficient for a TMLR publication, especially given there have already been many surveys mentioned by this study. The authors should incorporate more original contributions. Additionally, the work lacks experimental validation, such as comparative analyses of different approaches, which diminishes its practical impact.

---

> ### Author Response · Authors · 2025-03-11
> **Response**
>
> Thank you for recognizing the “comprehensive” nature and the “integrated view of the techniques” of our work, as well as noting that it provides “a good summary for readers to get familiar with this field.” Our primary goal is indeed to alleviate the burden on **systematically reviewing/understanding the technical intricacies of LIS frameworks in unified components**.
>
>
> ``merely summarizing and unifying others’ work is not sufficient for a TMLR publication, especially given there have already been many surveys mentioned by this study``
>
> We admit your concerns about originality under “experimental validation” and “new insights”.  We respectfully refer to TMLR’s acceptance criteria (https://jmlr.org/tmlr/acceptance-criteria.html), which ask:
>
> Are the claims made in the submission supported by accurate, convincing and clear evidence?
>
> Would some individuals in TMLR's audience be interested in the findings of this paper?
>
> In this context, as agreed by the other two reviewers, we believe the answers are both **"Yes"** regarding our SURVEY. Please note that previous acceptance on TMLR justifies that the experimental validation is not compulsory for publishing a SURVEY, e.g., https://openreview.net/pdf?id=wZLWuFHxt5
>
> We plan to incorporate EXPERIMENTAL VALIDATION in this survey only if they are deemed critical to make the paper's contribution (i.e., **allowing readers to systematically review/understand the technical intricacies of LIS frameworks in unified components**) **SOLID** and know where the experiments would lead us to.
>
> ``there have already been many surveys mentioned by this study``
>
> Existing surveys typically provide only a broad mention of LLM inference + search under a more general topic of LLM agents. In contrast, our survey offers a detailed, unified, and comprehensive perspective specifically focused on LLM inference + search . This addresses a critical unmet need. A similar situation is common in academic publishing—for example, although numerous LLM surveys published in 2023 covered reasoning works, top venues like EMNLP and ACL still accepted surveys specifically focused on LLM reasoning. The paragraphs below further refer to our paper and the comment from another reviewer to support the strong positioning of our work among other surveys:
> * As specified in Table 1, Section 1.1, they are far more enough to demonstrate the technical complication and the distinctions with other relevant directions (Section 6, 7) and provide a unified base to discuss their deviation, applicability, performance, efficiency and so on (Section 8).
> * As confirmed by Reviewer F53r, the previous surveys are "not dedicated surveys, with just have one related section" (or paragraph), let alone achieve the goal of providing an unified skeleton of looking into technical details of LIS frameworks. If there are any relevant works that we may have overlooked, we would be grateful if you could enlighten us.
>
> We believe the primary concern lies in meeting the **uniqueness** and **solidity** expected of a high-standard venue like TMLR. We respectfully argue that our survey meets these standards for the following reasons:
> 1. It presents a unified skeleton (high-level, mid-level and low-level modules in Figure 1) for organizing the field;
> 2. Within that skeleton, it abstracts different LLM-profiled roles (LMPRs) and analyzes distinct fine-grained instantiations;
> 3. It comprehensively incorporates high-quality, peer-reviewed works and presents them in a clear and structured manner with the help of the unified skeleton and LMPRs —an assessment supported by both of the other reviewers.
>
> **Question**
>
> Could you please provide further details on why both “Claims And Evidence” and “Audience” were selected as “No” in your review?  Happy to solve any **SPECIFIC** concerns.

---

### Review · Reviewer_G9WE · 2025-03-25

**Summary Of Contributions:**

This work conduct a survey for test-time computation, including tasks, LLM profiling, search algorithms, and relevant frameworks.

**Audience:**

Yes

**Claims And Evidence:**

Yes

**Requested Changes:**

* The Introduction part is too short. Please give more description in the introduction part.
* There is some red color text in the survey, while the text seems not special. Hence, this should be revised in the final version.

**Strengths And Weaknesses:**

* The survey is comprehensive and sufficient. This work takes a sufficient and comprehensive survey for the test-time computation. There are enough papers are introduced in this work.
* The paper is well-written with good structure.
* The paper contains many tables to help readers understand the different areas of test-time computation. For each Table, the authors cite sufficient papers and give detailed introductions so that it it easy to understand. For the important algorithms, the authors also gives very detailed descriptions.

---

> ### Author Response · Authors · 2025-03-26
> **Response**
>
> Thank you for your valuable feedback. We have addressed your comments as follows:
>
> * Converted the highlighted red text (previously used to mark changes) to standard formatting.
>
> * Expanded the Introduction to better clarify the strengths and limitations of LLMs, highlighting the need for search-based methods.
>
> We appreciate your suggestions and welcome any additional feedback.

---

### Decision · Action_Editor_6ziD · 2025-04-21

**Recommendation:** Accept as is

**Comment:**

Most reviewers find the survey comprehensive and sufficient for the specific topic of LLM inference via search.

One reviewer expressed concern about the narrower scope compared to broader topics such as general LLM inference or LLM agents. However, inference via search is a rapidly growing and important research area, and this survey offers a timely summary of relevant works for researchers new to the field.

Another main concern was that the submission merely summarizes others’ work and that there are already many surveys on the topic. It “lacks experimental validation, such as comparative analyses of different approaches”. Follow-up discussion clarified that this is a unique survey specifically focused on LLM inference via search. Furthermore, according to the TMLR Submission Guidelines, “surveys that … highlight trends … in an area” fall within scope.

Based on this reasoning, the submission meets the two acceptance criteria.

**Audience:**

This survey would be valuable for reviewers and researchers new to the topic of LLM inference via search, an important sub-area for scaling up LLM inference-time computation. It would also benefit LLM practitioners in selecting and implementing appropriate search algorithms for their specific use cases.

**Claims And Evidence:**

This submission presents a survey on the topic of LLM inference via Search (LIS). It identifies relevant works through a systematic search of recent publications from high-quality venues. The survey introduces a unified task definition based on a Markov Decision Process (MDP), provides modular definitions for LLM profiling and search procedures, and compares various inference frameworks. Reviewers are satisfied with the structure and consider it a comprehensive and sufficient overview of the topic. They find the detailed tables and diagrams helpful for readers. The discussion on related work highlights the survey’s unique focus on LLM inference via search, distinguishing it from other surveys on more broader topics of LLM agents.

---

> ### Author Response · Authors · 2025-04-25
> **Thanks**
>
> Thank you for your recommendation for acceptance. I truly appreciate your recognition—it means a great deal to me. As you noted, this work can benefit both "reviewers and researchers new to the topic" as well as "LLM practitioners," and I believe that dissemination through TMLR will effectively support this goal.